# Chronological age estimation from human microbiomes with transformer-based Robust Principal Component Analysis

Tyler Myers[1,2], Se Jin Song[1], Yang Chen[3,4,5], Britta De Pessemier [6], Lora Khatib[4,7], Daniel McDonald[4], Shi Huang [8], Richard Gallo[3], Chris Callewaert[6], Aki S. Havulinna [9,10], Leo Lahti [11], Guus Roeselers[12], Manolo Laiola[12], Sudarshan A. Shetty[12], Scott T. Kelley[13,14], Rob Knight[1,2,4,15,16] & Andrew Bartko [1,2,4] ✉

Deep learning for microbiome analysis has shown potential for understanding microbial communities and human phenotypes. Here, we propose an approach, Transformer-based Robust Principal Component Analysis(TRPCA), which leverages the strengths of transformer architectures and interpretability of Robust Principal Component Analysis. To investigate benefits of TRPCA over conventional machine learning models, we benchmarked performance on age prediction from three body sites(skin, oral, gut), with 16S rRNA gene amplicon(16S) and whole-genome sequencing(WGS) data. We demonstrated prediction of age from longitudinal samples and combined classification and regression tasks via multi-task learning(MTL). TRPCA improves age prediction accuracy from human microbiome samples, achieving the largest reduction in Mean Absolute Error for WGS skin (MAE: 8.03, 28% reduction) and 16S skin (MAE: 5.09, 14% reduction) samples, compared to conventional approaches. Additionally, TRPCA's MTL approach achieves an accuracy of 89% for birth country prediction across 5 countries, while improving age prediction from WGS stool samples. Notably, TRPCA uncovers a link between subject and error prediction through residual analysis for paired samples across sequencing method (16S/WGS) and body site(oral/gut). These findings highlight TRPCA's utility in improving age prediction while maintaining feature-level interpretability, and elucidating connections between individuals and microbiomes.

The human microbiome undergoes notable taxonomic and functional shifts as we age, influencing immune regulation, metabolism, dysbiosis, and disease risk[1,2]. In general, aging is accompanied by the loss of certain beneficial microbes and a rise in opportunistic ones[1]. For example, older adults tend to show reduced abundance of *Faecalibacterium prausnitzii* and related butyrate-producing gut bacteria, alongside expansion of pro-inflammatory taxa. On aged skin, sebaceous genera like *Cutibacterium acnes* decline, while *Corynebacterium* becomes more prominent[3,4]. In the oral cavity, age is associated with a drop in commensals such as *Streptococcus* and *Veillonella*, and an enrichment of anaerobes like *Fusobacterium*, *Treponema*, and *Porphyromonas* that are linked to periodontal disease[5,6]. Notably, these microbiome alterations correlate more with biological aging and frailty than with chronological age alone, indicating that a healthy aging microbiome is characterized by specific taxonomic patterns rather than just overall diversity or load. Chronological age simply counts years lived, whereas biological age, estimated from integrative molecular and clinical

biomarkers, more accurately reflects functional decline and health risk. Emerging microbiome clocks operationalize this concept by training machine-learning models on gut, skin, and oral community profiles to infer host age; the predicted value, often termed microbiome age, can deviate markedly from chronological age[7]. The magnitude and direction of this deviation correlate with phenotypes of healthy aging: older microbiome age ($\Delta$Age > 0) aligns with frailty and increased four-year mortality, whereas a younger-than-expected microbiome is linked to enhanced metabolic and inflammatory resilience[4,8–10].

In the gut, aging is typically marked by a shift away from a youth-associated community towards a dysbiotic profile. Studies in older adults consistently report a decline in core beneficial genera—including *Faecalibacterium*, *Coprococcus*, *Eubacterium*, and *Roseburia*—which produce anti-inflammatory short-chain fatty acids[1]. Meanwhile, potentially pathogenic or pro-inflammatory microbes increase with age: for instance, *Ruminococcus gnavus*, *Eggerthella lenta*, and various *Clostridium* species become more

abundant in the elderly and have been linked to frailty[1]. These compositional changes can promote chronic low-grade inflammation, impacting biological aging markers such as frailty indices and immune function. Although the persistence of a young-adult microbiome profile into old age appears detrimental, one study found that retaining high Bacteroides dominance in late life predicted lower 4-year survival, whereas a microbiome that shifts to a unique, less Bacteroides-centric composition was associated with healthier aging and longevity[9]. Conversely, certain bacteria often considered beneficial for metabolic health, like *Akkermansia muciniphila*, actually rise in abundance in many healthy seniors[1]. Exceptionally long-lived individuals tend to harbor higher gut microbial diversity and are enriched in taxa such as *Lactobacillus*, *Akkermansia*, *Methanobrevibacter*, and butyrate-producing *Clostridiales*, which may contribute to their reduced inflammatory status and improved metabolic profiles[11]. Altogether, a gut microbiome that maintains keystone anti-inflammatory species and limits overgrowth of pathobionts is associated with healthier aging, whereas the opposite pattern correlates with accelerated biological aging and age-related diseases.

The oral microbiome also shifts discernibly with age, with implications for both oral and systemic health. Older adults typically experience a decline in dominant health-associated genera like *Streptococcus*, *Granulicatella*, and *Veillonella*, along with an increase in anaerobic taxa implicated in periodontitis and oral dysbiosis[5,6]. For example, higher relative abundances of *Fusobacterium*, *Treponema*, and *Porphyromonas* are observed in older individuals, reflecting a microbiota more prone to gum disease and chronic inflammation[5]. This age-related oral dysbiosis can exacerbate periodontal disease, tooth loss, and even contribute to systemic inflammation, as oral pathogens have been linked to conditions like atherosclerosis and aspiration pneumonia. Importantly, lifestyle and socioeconomic factors modulate the aging oral microbiome. Cigarette smoking causes a broad shift in oral bacterial communities, with active smokers show significantly lower levels of *Proteobacteria* (e.g., *Haemophilus* spp.) and commensals such as *Capnocytophaga*, alongside enrichment of others like *Streptococcus*[12]. Such changes compound with age, potentially worsening oral health outcomes. Similarly, differences in diet and oral hygiene associated with socioeconomic status can lead to distinct oral microbial profiles in older populations, partly accounting for health disparities[13]. Notably, measures of oral microbiome diversity tend to decrease with frailty even as some diversity indices increase with age[6], suggesting that a balanced oral microbiota is a marker of healthy aging. Taken together, maintaining an oral microbiome dominated by beneficial, inflammation-mitigating species may help mitigate age-related oral diseases and their ripple effects on overall health[12].

The skin microbiome also changes with age, impacting skin health and aging processes[14,15]. Age-related alterations in the skin microbiome are influenced by both intrinsic and extrinsic factors, such as smoking habits and ethnicity, shedding light on the complex interplay between the host and microbial communities on the skin[16]. The skin microbiome has been associated with signs of aging, and skin diseases such as acne and atopic dermatitis, highlighting its role in maintaining skin homeostasis and health throughout the aging process[17–19]. Differential abundance analysis of skin microbiomes distinguishes chronological age from signs of aging, uncovering markers associated with greater grade of wrinkles, transepidermal water loss, and loss of skin hydration[18]. The ability to infer signs of aging and chronological age based on the composition of the human skin, oral, and gut microbiomes underscores the potential for microbiome-based interventions in promoting healthy aging, and opportunities to improve microbiome data analysis using advanced machine learning (ML) and deep learning methods[20].

Machine learning has become integral in analyzing microbiome data, enabling prediction of host characteristics like age from complex community profiles. Early approaches typically used taxonomic features (e.g., relative abundances of 16S rRNA amplicon sequence variants) and traditional classifiers. For example, a 2020 multi-cohort study applied random forests to 16S data and found that microbiomes can predict chronological age from skin microbiomes within 3.5 years on average[20]. Notably, both

taxonomic composition and functional profiles (metabolic pathways from metagenomes) provide predictive signal. In infants, the gut microbiome's species composition and metagenomic pathways were highly predictive of developmental age ($R^2 \approx 0.7$–$0.8$, Mean Absolute Error ~1.4 months)[21]. This demonstrates that microbial community structure and function each capture age-related changes.

Random Forest (RF) models have been a benchmark for microbiome-based age prediction. RFs handle feature abundance data well and offer robustness to noise. In a large study spanning gut, oral, and skin microbiomes, an RF model achieved a MAE of ~3.8 years for skin microbiome age prediction, outperforming oral (4.5 years) and gut (~11.5 years) predictions[20]. This indicated that the skin microbiome can be a particularly strong clock for chronological age in adults. These studies established RFs on 16S or metagenomic features as a reliable baseline, while highlighting body-site differences (skin microbiome often showing the greatest age correlation).

Deep neural network models have been introduced to capture complex, non-linear patterns in microbiomes. DeepMicro, is one such model using an ensemble of autoencoder-based pipelines to classify phenotypes from microbiome data[22]. Autoencoder models learn a compressed, denoised representation of the data, isolating the most relevant features for representing the sample. Although DeepMicro was developed for disease prediction, its architecture (unsupervised feature extraction with supervised learning) has influenced microbiome age prediction efforts. Other deep models incorporate microbial phylogeny or time-series structure. For example, phyLoSTM is a deep framework that combines a convolutional neural network (CNN) with a long short-term memory network (LSTM) to analyze longitudinal microbiome data[23]. In a 2021 study, phyLoSTM was applied to time-course gut microbiome profiles and outperformed RF by 5–19% in AUC for health outcome prediction[23]. This CNN–LSTM approach extracts local taxonomic patterns via convolution and captures temporal dependencies via recurrent units, which could be adapted to improve age predictions in longitudinal aging cohorts (e.g., tracking an individual's microbiome age trajectory). Similarly, the MDeep method leverages phylogenetic tree depth and deep learning to outperform competing methods for regression and binary classification[24]. The hierarchical taxonomic convolutional layers mirror taxonomic levels in the phylogenetic tree, allowing for an informative compression of OTUs before passing the representation to a fully connected, dense neural network– with the assumption that phylogenetically adjacent OTUs effect the target prediction in the same direction (i.e., one cluster of adjacent OTUs would be assumed to be only associated with older individuals). MDeep outperforming other neural network architectures, as well as Random Forest, demonstrates the importance of contextualizing the microbiome for improving model performance.

Microbiome clock models have emerged as specialized regressors for host age. These often integrate multiple data types and algorithms to boost accuracy. Initializing these efforts, in combination with deep learning approaches, an aging clock was demonstrated as the best approach for predicting host age from gut microbiome profiles, with a MAE of 5.9 years and providing a starting point for anti-aging intervention via feature importance[7]. Furthermore, the analysis provided foundations for feature extraction from deep learning models by evaluating the shift in model performance as a function of feature value perturbation[7]. Another example of leveraging machine learning for age prediction was an ensemble model developed using "multi-view" input, combining species abundance and pathway abundance profiles from gut metagenomes[25]. The integrated model (an ensemble of heterogeneous learners) achieved high accuracy in predicting chronological age ($R^2 \approx 0.60$, MAE ~ 8.3 years) after accounting for geographic and technical confounders[25]. The inclusion of functional data alongside taxonomy improved performance, underscoring that both community composition and its functional gene content change with age. Another advanced model called gAge introduced a composite biological age predictor based on the gut microbiota[8]. The gAge framework integrates gut microbial features "from different perspectives" (e.g., taxonomic, functional,

**Fig. 1 | Preprocessing and model overview to transformer-based RPCA.** Samples represented as count tables are visualized and converted to RPCA vectors. RPCA vectors are input as sequences into a transformer encoder model with multi-head attention. The transformer model outputs are provided to a classification (CLS) or regression (REG) head for classification, regression, or Multi-task learning.

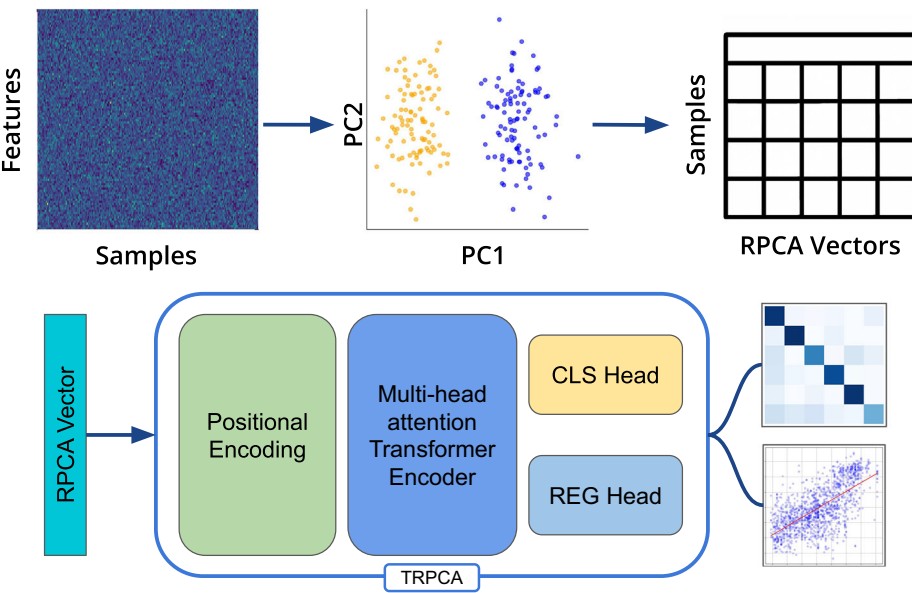

and possibly metadata) using an ensemble of models, and notably it leverages unpaired samples to use more data[8]. This ensemble significantly improved prediction accuracy over any single model[8]. In an elderly cohort, gAge's output and the deviation of predicted age from actual age (the residual) were strongly associated with health and frailty status[8]. This suggests microbiome-based age predictors can serve as clocks for biological age, where a microbiome "older" or "younger" than expected correlates with frailty or healthy aging, respectively.

Several studies have applied these models to various body sites and populations. Apart from adult gut clocks, researchers have built age predictors for early life and other niches. A XGBoost model was trained on healthy infant gut microbiomes to define a "microbiota age," and a derived z-score was used to identify growth faltering[21]. In that work, specific bacteria (e.g., *Faecalibacterium* spp., *Bifidobacterium* spp.) and microbial pathways were identified as top contributors to age predictions, reflecting the orderly succession of the gut microbiome in infancy[21]. On the other end of the spectrum, studies of older adults indicate that chronological age itself may be less predictive within a narrow range of elderly years and, instead, microbial shifts often align more with biological aging or frailty. For instance, a longitudinal analysis found that microbiome composition in older adults tracked with frailty more than chronological age differences[4]. Together, these benchmark studies showcase the range of models (from random forests to deep neural networks) and use-cases (gut, oral, skin, infant, elderly) in microbiome age prediction, along with typical performance metrics. Currently, gut microbiome-based clocks for human chronological age can achieve roughly 5–10 year accuracy in adults[25], while specialized contexts (like infant growth or multi-omics clocks) can be even more precise[21,26].

Another emerging theme is the application of transformer-based architectures and large-scale pretraining for microbiome analytics[27]. Transformer neural networks, which excel at modeling sequence and context due to the attention mechanism, are being adapted to microbial data to capture complex relationships across hundreds of taxa or genes. Transformer models can excel at microbiome analysis because they can attend to multiple microbial species simultaneously, identifying both obvious and hidden relationships between different taxa that might influence health outcomes, much like how understanding a conversation requires recognizing connections between words spoken at different times. Recent reviews highlight that transformer models can create rich, context-aware embeddings of microbiome profiles that outperform traditional models on downstream tasks[28]. For example, the MetaTransformer model was used to integrate metagenomic and metabolomic data, yielding improved predictive

power in microbiome studies[28]. Transformers can attend to subtle co-occurrence patterns and long-range interactions (much like learning a language of microbial communities), which is promising for improving age predictions from high-dimensional metagenomic data. While still in early stages, such deep learning innovations are likely to enhance microbiome-based age predictors.

Here, we propose a transformer-based approach where vectors from Robust Principal Component Analysis (RPCA) are treated as sequences representing each microbiome sample[29]. These RPCA sequences provide an appropriate sample representation for the transformer model, allowing the attention mechanism of the model to train over the high dimensional principal component space, while simultaneously accounting for the features that explain the variance of the principal component. Our model differs from previous deep learning approaches to microbiome analysis by applying transformer architecture and attention mechanisms (without pretraining) to learn patterns between orthogonal components derived from RPCA dimensionality reduction. This approach enables our model to capture contextual relationships within microbiome samples, similar to MDeep and phyLoSTM frameworks, while offering key architectural advantages: it eliminates overfitting risks during data compression and allows phylogenetically similar taxa to exhibit different feature importance directionalities. By assembling a comprehensive investigation of microbiome and age, with 16S rRNA sequence data from 8959 samples from 10 studies and metagenomic data from 9356 samples from 56 studies, we demonstrate that TRPCA is at least comparable to other popular machine learning models for all skin, oral, and gut microbiomes tested, regardless of sequencing method, while improving performance when trained via MTL. Furthermore, we describe methods to interpret the features from TRPCA on a global and sample level using SHapley Additive exPlanations (SHAP) values and RPCA feature loadings, leveraging the TRPCA architecture to compare the accuracy of age prediction across body site and sequencing method, uncover features driving the process of aging, and highlight the age prediction residual as a reproducible host associated attribute.

## Results and discussion
### TRPCA improves age prediction from gut, oral, and skin microbiome profiles

TRPCA (Fig. 1) improved MAE for host age prediction by up to 28% compared to the other model architectures (Support Vector Regression (SVR), Gradient Boosting Regression (GBR), K-Neighbors Regression (KNN), Neural Network Regression (NN), and RF), demonstrating the

**Table 1 | Comparison of the MAE from the best performing model and TRPCA**

| Target | Sample Type | MAE (Model) | MAE (TRPCA) | R2 (TRPCA) | $n$ | %improvement | Study |
|---|---|---|---|---|---|---|---|
| Age | Skin (16S) | 5.73 ± 1.25 (KNN) | 5.09 ± 1.07 | 0.56 ± 0.19 | 1975 | 14 ± 6.7 | 20 |
| Age | Oral (16S) | 7.47 ± 2.05 (SVR) | 7.02 ± 1.82 | 0.42 ± 0.11 | 2550 | 6 ± 10.5 | 20 |
| Age | Gut (16S) | 11.66 ± 0.36 (SVR) | 11.42 ± 0.55 | 0.17 ± 0.08 | 4434 | 2 ± 1.9 | 20 |
| Age | Skin (WGS) | 11.14 ± 2.65 (KNN) | 8.03 ± 4.10 | 0.54 ± 0.31 | 115 | 28 ± 20.3 | 69 |
| Age | Oral (WGS) | 7.16 ± 0.69 (KNN) | 6.72 ± 0.81 | 0.45 ± 0.13 | 600 | 6 ± 6.1 | 69 |
| Age | Gut (WGS) | 9.98 ± 0.8 (KNN) | 8.83 ± 0.50 | 0.71 ± 0.04 | 8641 | 12 ± 2.7 | 69 |

*MAE* Mean absolute error, *KNN* K-nearest neighbor regressor, *SVR* Support vector regressor, *GBR* Gradient boosting regressor, *RF* Random forest regressor.

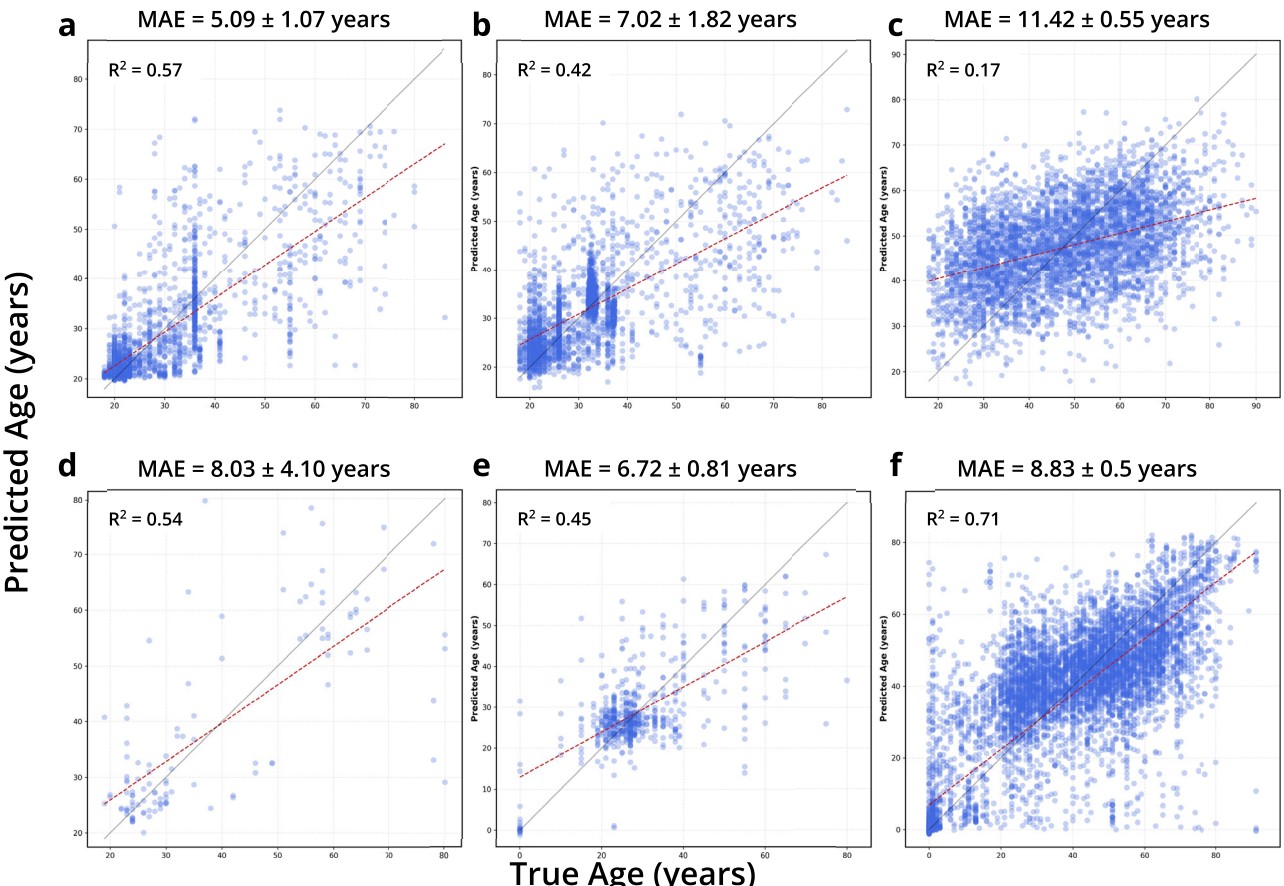

**Fig. 2 | Regression for age prediction by sequencing method and body site. a–c** are regressions for 16S data and **d–f** for WGS data. **a, d, b, e, c, f** indicate regressions for skin, oral, and gut microbiome samples respectively.

greatest improvements in MAE for the skin microbiome samples (Table 1, Fig. 2, Supplemental Data 1). Although TRPCA improved age prediction across datasets when grouping samples by subject ID, our analyses showed inferior age prediction for 16S skin and oral microbiomes compared to RF models from Huang et al., on the same datasets. To determine whether the discrepancy was due the difference in how the train/test splits were grouped and stratified, TRPCA models were benchmarked with Huang et al. and evaluated for only 16S skin and oral samples following the exact same stratification criteria for samples denoted in Huang et al. (without grouping by subject ID). The TRPCA model had an average MAE of 2.16 years for 16S skin samples and 3.86 years for 16S oral samples (Fig. S1), compared to the reported 3.8 years and 4.5 years, respectively. To further investigate the influence of samples from repeated measures in the 16S skin and oral data, we created two filtered datasets. One included the top 40 subjects with at least two samples stratified by subject ID, and the other included only one

sample per subject. Using the filtered datasets, TRPCA outperformed the other models for host age prediction for the filtered datasets, suggesting that including multiple samples from an individual did not explain the TRPCA model's advantage over other techniques (Figs. S2–S3). When trained on the dataset including multiple samples per host, TRPCA predicted host age with a MAE of 0.61 years for 16S skin samples and 0.42 years for 16S oral samples (Fig. S2), outperforming the RF model with a MAE of 1.84 years for 16S skin samples and 1.31 years for 16S oral samples in this scenario. Finally, we investigated the performance of TRPCA compared to the other model architectures on the 16S skin dataset with varying sample sizes. As anticipated for deep learning models, TRPCA performed better than conventional models for larger sample sizes when predicting host age; however, TRPCA achieved similar performance to conventional models even for smaller sample sizes (Fig. S4), demonstrating its utility for small datasets as well.

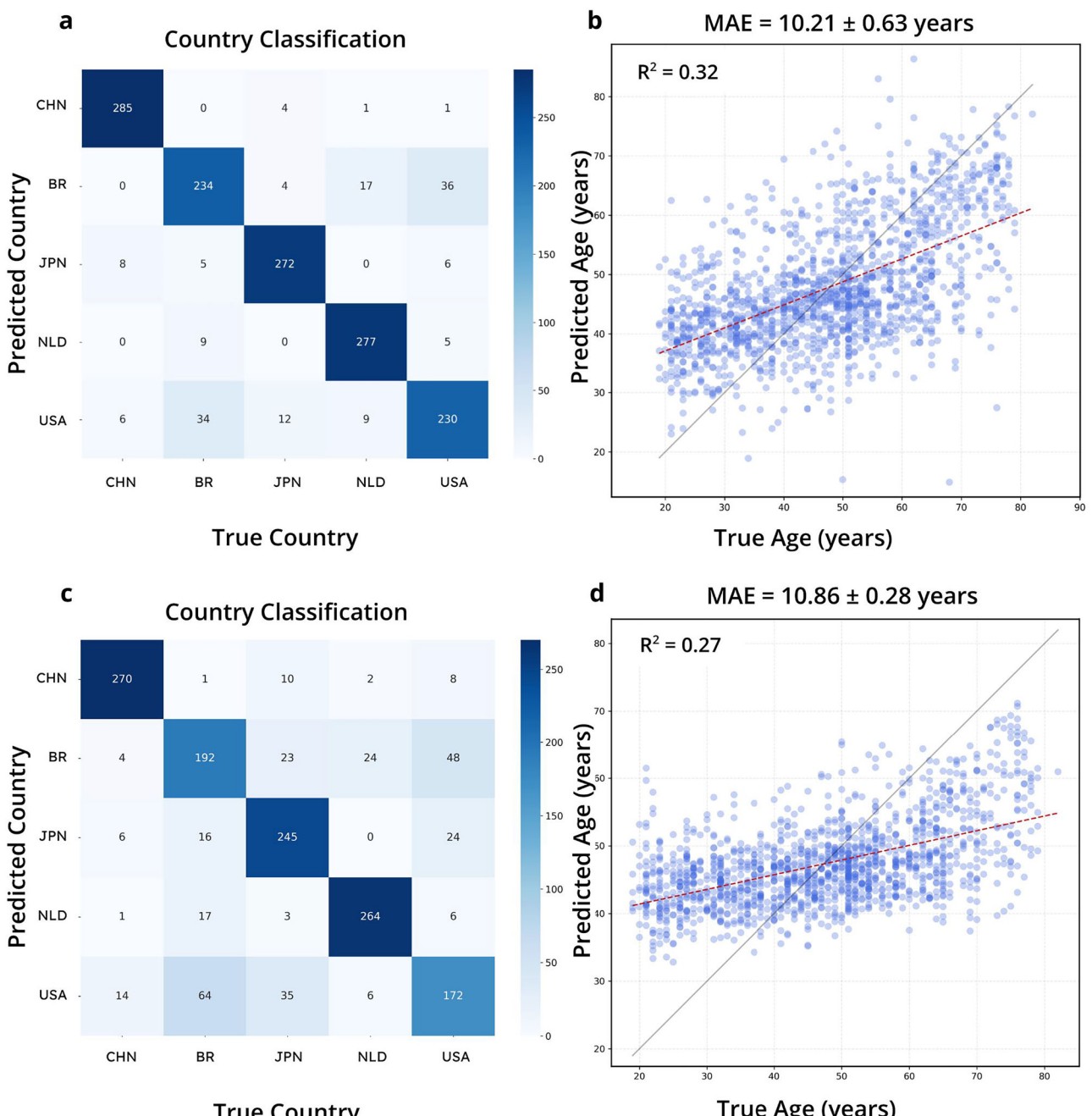

**Fig. 3 | MTL performance on country of birth classification and age prediction for WGS Stool samples from curatedMetagenomicData. a, b** are the confusion matrix and age prediction regression for the TRPCA MTL model. **c, d** are RF models trained individually for the same classification and regression tasks as the TRPCA MTL model.

## MTL for adult WGS gut samples classifies country of birth and predicts host age

For the combined task of classifying host birth county and predicting host age, we took 270 samples from each of five countries (China:CHN, United Kingdom:BR, Japan:JPN, Netherlands:NLD, United States of America:USA) in the WGS adult gut dataset. To benchmark the conventional models, we trained the models for classification and regression models separately. Then, we selected the top-performing model for each task to act as the ensemble model for comparison with TRPCA. TRPCA achieved a MAE of 10.21 years for host age prediction and an accuracy of 0.92 for country of birth prediction, compared to the best MAE of 10.90 years using RF and the best accuracy of 0.80 using RF (Fig. 3, Table 2). TRPCA improved host age prediction by 6% and birth country classification by 13% improvement, compared to RF.

## Interpretable feature selection from TRPCA via SHAP values

After training each model for regression tasks, SHAP values were calculated on the test split of the data following prediction from the regression tasks. From the SHAP values matrix and RPCA feature loading matrix, global mean importances (Figs. S5–S10) and sample level importances (Fig. 4, Figs. S11–S15) were calculated. Global mean importances were displayed as differentials to highlight features identified for host age, with features of positive mean importance being associated with older subjects and features of negative mean importance being associated with younger individuals (Figs. S5–S10). SHAP values on the sample level were shown using a normalized heatmap, highlighting the sample level features identified by the TRPCA model for age prediction. The top 50 features identified by TRPCA for 16S skin were prevalent across the majority of test samples and showed clustering

**Table 2 | MTL performance**

| Target | Sample Type | Metric (Model) | Metric (TRPCA) | n | %improvement | Study |
|---|---|---|---|---|---|---|
| Age | Gut (WGS) | 10.86 ± 0.48 (RF) | 10.21 ± 0.63 | 1350 | 6 ± 1.7 | 69 |
| Birth Country | Gut (WGS) | 0.79 ± 0.04 (RF) | 0.89 ± 0.02 | 1350 | 13 ± 3.1 | 69 |

Age prediction and Birth Country classification for the top 5 countries from CuratedMetagenomicData evenly sampled for TRPCA MTL model vs best alternative model. Metric for regression and classification are MAE and Accuracy, respectively. TRPCA MTL model was trained simultaneously on both the classification and regression tasks.

patterns on a sample and feature basis (Fig. 4). The sample level interpretation of the SHAP values provided a high-resolution view of the feature importances, highlighting features that may be prevalent for most samples, as well as features that may only be important for the age prediction of a subset of the data. The abundance of *Enterococcus faecalis*, for example, was a globally significant feature, because most samples had a normalized feature importance greater than 0.4 for this feature (Fig. S12). However, for the first cluster of samples, *E. faecalis* abundance was the most important feature for age prediction, and other features had low feature importances. Global feature importances were calculated by averaging the feature importances for each sample. For the top global features, correlations were calculated between the log-transformed values for the feature versus host age (Figs. S16–S21). To further validate the identified features from TRPCA, we explored the individual contributions of features for WGS skin samples (Fig. 5). Using the sample level feature importances derived from TRPCA, we were able to hone in on the individual contribution of a single feature for the age prediction of a given sample (Fig. 5a). Negative feature importances(colored in blue) indicate that the feature influenced the model prediction towards a younger prediction for that individual, whereas positive feature importances(colored in red) drove the model prediction towards an older age. For example, for the age group 28–35, we see individuals with compositions of *Cutibacterium granulosum* and *Malassezia globosa* which drive the models predictions towards a younger prediction. Lastly, to investigate the derived feature importances from TRPCA, we compared the feature importances from each model in a pairwise manner using Pearson correlation (Fig. 5c) and plotted normalized feature importances between TRPCA and RF for visualization (Fig. 5b, Fig. S22). TRPCA derived features show strong agreement with SVR, KNN, GBR, and RF models; however, TRPCA, SVR, KNN, GBR, and RF derived features demonstrated significant divergence from the NN predictions, with the NN consistently underperforming for evaluations.

**Paired microbiome samples exhibit consistency in age prediction error residuals**

To further explore the differences between 16S and WGS data types, we trained TRPCA on paired 16S and WGS samples from the THDMI and FINRISK cohorts used for effect size comparisons in Greengenes2, where each subject has a corresponding 16S sample for each WGS sample[30]. Models were trained using the default parameters and ten-fold cross-validation (Fig. 6). Age prediction for WGS showed a MAE of 8.68 years, compared to a MAE of 9.85 for 16S (WGS samples for this analysis were processed using methods described in Greengenes2, rather than the curatedMetagenomicData pipeline). Although the host age MAEs differed significantly, we investigated the correlation between the prediction error residuals for each sample. In line with prior effect size consistency reported with 16S and WGS data, we report a high degree of correlation in the age residuals from 16S and WGS prediction ($R^2 = 0.63$)[30]. To determine whether samples from different body sites from the same subject exhibited a similar pattern, we selected WGS samples from curatedMetagenomicData where individuals had both gut and oral microbiome samples. The resulting dataset had 242 subjects, with one sample per body site. We ran TRPCA with default parameters on the dataset with a Cross validation(CV) = 10, stratifying the data splits by study. The resulting regressions showed no significant difference for age prediction between the oral (MAE = 9.11) and gut samples (MAE = 9.5) for the 242 subjects (Fig. 6). As for the age

residuals, we found a moderate correlation between the age prediction residuals for paired WGS oral and WGS stool samples ($R^2 = 0.34$).

**Skin microbiome**

TRPCA was able to identify several key bacterial species that have already been shown to be associated with skin health, disease, and aging[31], including *Corynebacterium, Lactobacillus, Cutibacterium acnes*, and *Staphylococcus*. *Corynebacterium simulans* has been shown to be more abundant in adult skin than childhood skin, indicating a link with sebum secretion and a potential role in the aging process[32]. *Lactobacillus*, identified as a feature associated with younger individuals, has been linked to skin health through antimicrobial activity against skin pathogens, highlighting their potential role in maintaining skin health and reducing skin inflammation[32–34]. *Cutibacterium* is also crucial for skin health, showing correlations with chronological aging and signs of aging[18]. Specifically, *Cutibacterium acnes* is a dominant bacterium on human sebaceous skin, and particular clades are commonly associated with acne vulgaris[35]. *Cutibacterium* is most dominant during puberty on sebaceous skin sites (including face, scalp, upper back), and decreases in abundance as sebum secretions decrease with older age[36]. Its higher abundance in younger individuals makes it a well-known marker for host age[37,38]. However, its over-proliferation has been implicated in contributing to acne, and specific strains of *Cutibacterium acnes* dominate the skin microbiome of adult acne patients, indicating that its involvement in skin conditions may complicate the age signal[38,39]. *Cutibacterium acnes* metabolic products, such as short chain fatty acids, act on the epidermis to promote lipid synthesis and improve barrier function[40]. With TRPCA, we found that *Cutibacterium* has a negative correlation with age from 18 years onwards, suggesting that the age signal is dominant over acne signals in the general population (Figs. S20 and S21). On the other hand, *Staphylococcus epidermidis*, another prevalent skin microbe contributing to skin health and infections, showed a positive correlation with age (Figs. S20 and S21). *Staphylococcus epidermidis* enables *Cutibacterium acnes* to form biofilms under certain conditions[41]. *Staphylococcus aureus*, a major human pathogen, as well as *Staphylococcus epidermidis*, a commensal, have important roles in skin diseases whereas coagulase negative *Staphylococci*, such as *Staphylococcus hominis*, protect the skin[42–44]. *S. epidermidis* has a positive correlation with age as of 18 years old and up (Figs. S20 and S21).

**Oral microbiome**

Key bacterial taxa for TPRCA age prediction for oral microbiome samples such as *Actinomyces, Fusobacterium, Neisseria, Veillonella, Rothia mucilaginosa*, and *Prevotella* have been previously identified for their significance in the oral microbiome and its relationship with aging. *Actinomyces* is a prevalent genus in the oral cavity associated with oral health[45]. *Fusobacterium* has been linked to various oral health and systemic health conditions including periodontitis, acute pancreatitis and cancer, and *Fusobacterium* abundance may increase with age and impact oral health outcomes[46,47]. *Neisseria* is associated with the healthy core microbiome of the human oral cavity and plays a role in maintaining oral health, and patients with more Neisseria have improved patient outcomes in the context of adenoid cystic carcinoma and head and neck squamous cell carcinoma[48,49]. *Veillonella* and *Rothia mucilaginosa* may influence oral health outcomes, aging, and blood pressure, by modulating nitric oxide bioavailability, with *Veillonella* appearing as a genus to decrease in abundance with age in our analysis[50]. *Prevotella* species may help maintain oral health and be influenced by aging, HIV status, and country of birth[49,51].

# 16S Skin Feature Importance

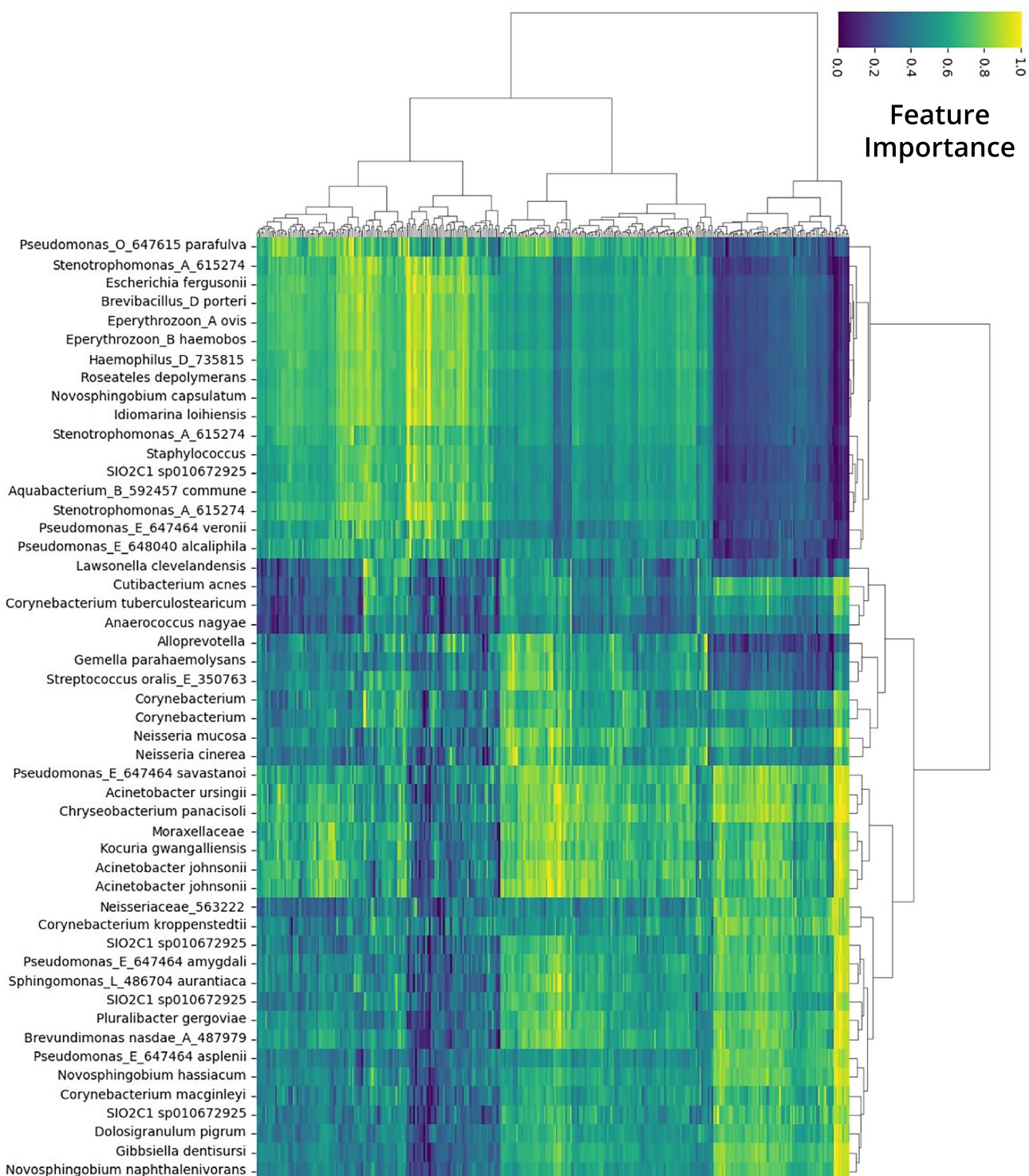

**Fig. 4 | Top 50 normalized and clustered feature importances for 16S Skin microbiome features.** The feature importances are derived from the dot product of the PCA and SHAP matrices with rows as samples and columns as assigned taxonomy from influential ASVs. Redundant taxonomy values are indicative of AVSs which map to the same taxonomy classification. Clustering of columns indicate influential taxa groupings for age prediction, whereas clustering of rows highlight individuals with similar significant features.

## Gut microbiome

Among the taxa identified by TRPCA, *Akkermansia muciniphila*, *Faecalibacterium prausnitzii*, *Bacteroides*, *Bifidobacterium*, and *Roseburia* have been extensively studied for their roles in influencing health outcomes in aging individuals. *Akkermansia muciniphila*, a mucin-degrading bacterium, has been associated with various health benefits, including metabolic health and anti-inflammatory effects. In agreement with the TRPCA feature importance, studies have shown that *Akkermansia muciniphila* levels were

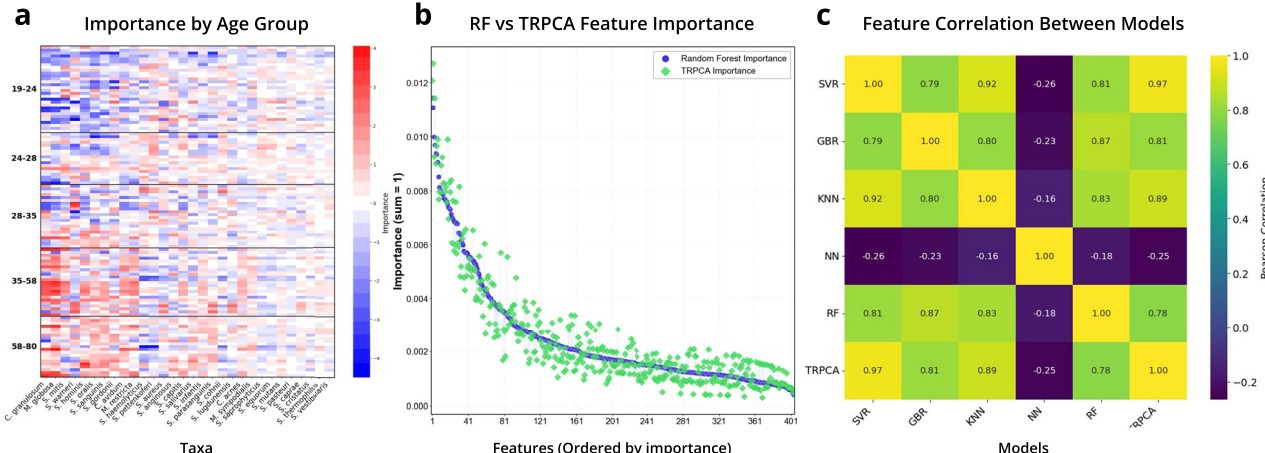

**Fig. 5 | Interpretable feature importances. a** Sample level feature importances for WGS Skin microbiome samples. Features colored in blue indicate a feature that influences the prediction of that sample to be younger, whereas red is indicative of a feature that drives the prediction to be older. **b** Feature level comparison of feature importances for WGS skin between TRPCA and RF model. **c** A pairwise comparison of the Pearson correlation between feature importances for each model architecture. SVR, GBR, KNN, RF, and TRPCA feature importances are highly correlated for WGS Skin microbiome feature importances.

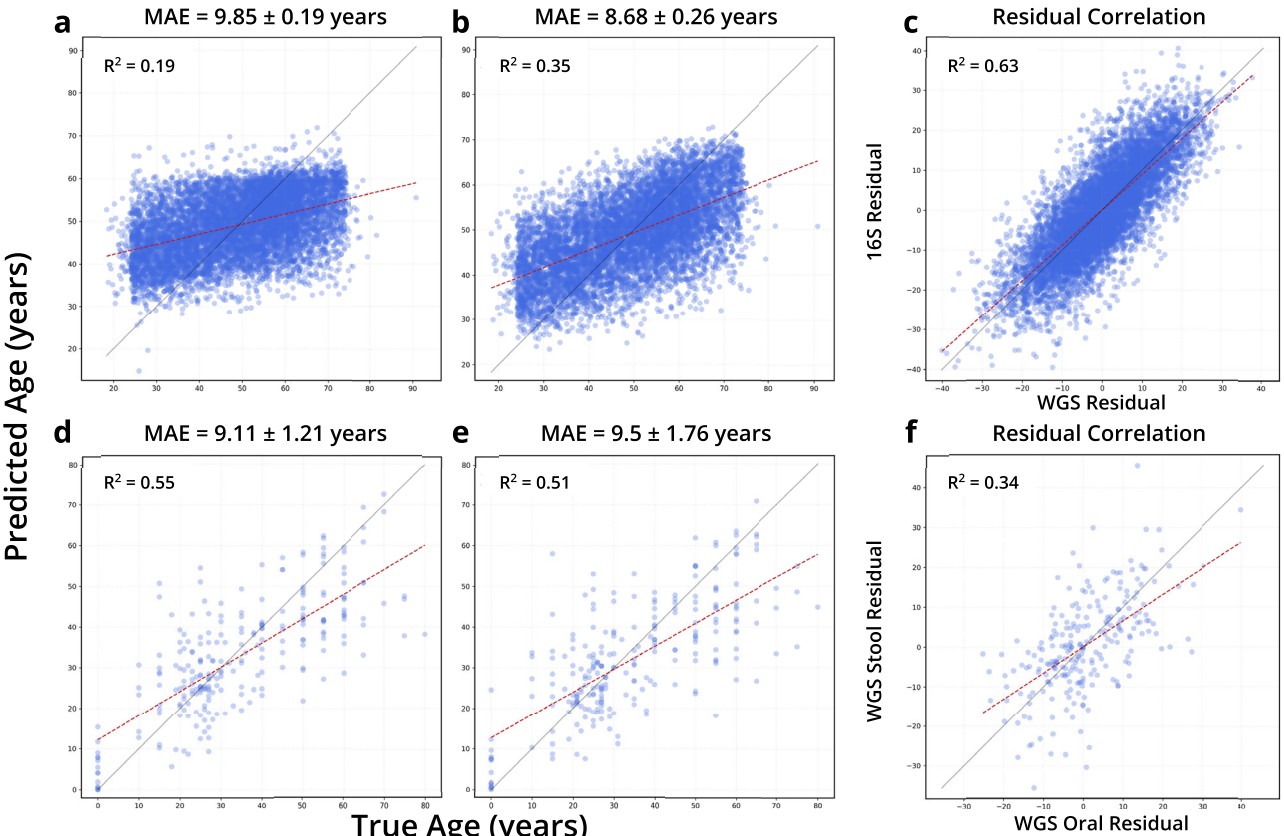

**Fig. 6 | Paired sample age residual errors. a** TRPCA predictions for 16S stool samples from the THDMI and FINRISK cohorts. **b** TRPCA predictions for WGS stool samples from the THDMI and FINRISK cohorts. **c** The correlation between prediction errors in paired 16S and WGS stool microbiome samples (R2 = 0.632). **d** TRPCA predictions for WGS oral microbiome samples from curatedMetagenomicData. **e** TRPCA predictions for WGS stool microbiome samples from curatedMetagenomicData. **f** The correlation between prediction errors in paired WGS oral and WGS stool microbiome samples (R2 = 0.339). The error in model prediction for paired samples (WGS Stool/16S Stool and WGS Stool/WGS oral) implies host associated attributes may be associated with residual error.

increased in the control group compared to individuals with specific health conditions, as well as older individuals, indicating a potential protective role in healthy aging[52–54]. *Faecalibacterium prausnitzii*, known for its butyrate-producing capabilities, has been linked to gut health and immune modulation. *Faecalibacterium prausnitzii* abundance is negatively correlated with circulating branched-chain amino acids and insulin resistance, highlighting its potential metabolic implications[55–59]. As identified by TRPCA as a top feature for 16S stool samples, *F. prausnitzi* abundance decreases with age, suggesting implications in aging[1]. Positive correlations between *Bacteroides* and age have been reported, suggesting a potential role in the rate of aging[60,61].

*Bifidobacterium* levels have been negatively associated with age and disease severity, indicating its potential impact on health outcomes[62,63]. Decreased levels of *Roseburia* have been observed in conditions associated with dysbiosis, suggesting a possible role in maintaining gut homeostasis[62,64,65].

## Biological and technical insights from residual age predictions

Paired WGS microbiome samples from different body sites showed only a moderate alignment in predicted aging. Specifically, we observed a modest correlation ($R^2 \approx 0.34$) between age predictions from oral versus stool WGS microbiomes. This level of concordance likely reflects fundamental microbiome body-site differences. The oral and gut communities are compositionally distinct, with over 90% of taxa in each habitat are unique and not shared with the other, limiting cross-site prediction agreement[66]. Prior studies confirm that microbiome aging signatures are highly site-specific: for example, the skin microbiome can predict age much more accurately than the gut microbiome, potentially obfuscating the correlation strength between the paired residuals[20]. Thus, an individual's oral and gut microbiotas may each track aspects of host aging, but differences in local environment, diet, and host factors at each site dampen the correlation. Nonetheless, the moderate correlation underscores that chronological age imprints on each body-site microbiome in a distinct manner, in line with extensive evidence that the microbiome is compartmentalized[66].

In contrast, we found a much stronger residual correlation ($R^2 \approx 0.63$) between 16S rRNA gene versus WGS age predictions for the paired stool samples. This high correspondence indicates that when one sequencing modality's model over- or under-predicted a person's age, the other tended to show a similar residual. Several factors may explain this alignment of residuals. First, it could reflect shared biological signals captured by both methods: the individuals who are hard to predict (or biologically unusual for their age) via 16S are likewise outliers via WGS. In other words, method-independent aging features drive the prediction errors in both cases. This is a promising scenario, suggesting our models are detecting real host-microbiome age relationships rather than random noise. Alternatively, systematic biases in sample processing or subject selection might influence both 16S and WGS similarly. For instance, if certain subpopulations cause microbiome age to deviate from chronological age, both sequencing methods could identify these deviations. We also consider the role of taxonomic resolution with shotgun sequencing providing finer species level detail absent in 16S profiling. In theory this should improve accuracy, and indeed shotgun models often modestly outperform 16S[67]. However, the high residual correlation implies that, although the age prediction accuracy with WGS data was lower due to improved species resolution, the associated deviation in model prediction remains consistent, posing the age prediction residual as a reproducible measure. Such method-independent agreement supports the idea that the residuals reflect true biological variation in aging pace, as opposed to method-specific artifacts. In summary, while different sequencing technologies have inherent biases, the convergence of their age predictions in our study suggests we have identified robust aging signatures in the gut microbiome that transcend technical platforms. This is encouraging for future efforts to develop microbiome-based age biomarkers, as it indicates a person's microbiome age is not merely an artifact of the sequencing method but rather grounded in biological signals.

Our analysis of model feature importance by age group revealed that certain microbial features disproportionately influenced model predictions, notably in individuals aged 28–35 for WGS skin data (Fig. 5a). In this younger adult cohort, the model tended to predict lower ages as a function of a feature's abundance. SHAP values highlighted two organisms in particular, *Cutibacterium granulosum* and *Malassezia globosa*, as drivers of skewed age estimates. By examining SHAP values, we identified concrete microbial candidates that may serve as sentinels of accelerated or decelerated microbiome aging in an individual based on the influence of the feature to drive model prediction toward younger or older ages.

Beyond these two taxa, our models identified a broader panel of age-associated microbes across skin, gut, and oral sites, each with potential impacts on host health. For instance, a relative abundance of certain oral anaerobes (e.g., *Fusobacterium*) emerged as markers of an older oral microbiome in our models, consistent with literature showing periodontal bacteria increase with age in some individuals[6]. These organisms are well-known contributors to gum disease, which tends to worsen with age and can have systemic inflammatory effects. Their prominence in age-prediction models suggests that oral health status is reflected in the microbiome age signal. Likewise, in the gut we noted that model coefficients and SHAP values often flagged depletion of beneficial genera (e.g., *Faecalibacterium*) and enrichment of pro-inflammatory taxa (e.g., *Escherichia/Shigella*) in individuals predicted to be older than their years. The ability to predict host age from microbiome data carries several translational implications. In the realm of preventive and personalized medicine, a microbiome age model could be developed into a biological age indicator alongside other clocks (epigenetic, proteomic, etc.). There are also clear opportunities in the skincare and cosmetic industry. The skin microbiome was a strong component of our models, and prior work shows it to be a reliable mirror of chronological age[18,20]. Probiotic supplementation is another direction as our findings identify specific taxa (*Faecalibacterium* in the gut, or *Veillonella* in the oral cavity) that decline with age[6]. Restoring fiber-degrading, butyrate-producing bacteria in an elderly gut microbiome might not only make the microbiome profile younger but also confer benefits like improved colonic health, reduced inflammation, and better metabolic regulation[68].

Our microbiome aging models demonstrate performance on par with, or exceeding, previously published microbial aging clocks. For example, Galkin et al. (2020) reported a deep learning model for gut microbiome age prediction with a MAE of ~5.9 years on an independent test set[7]. The finding in this study is significant, as the model performed well on data that was uncharacteristic and out of the training distribution, as well as on subjects that were donors with type 1 diabetes; however, as neural networks are continuous function approximators, well-regularized models can generalize on unseen data. It should also be noted that the MAE metric of 5.9 years is based on a dataset with multiple samples from a given individual, and with a median age assignment error of 9.27 years. To give context for our WGS stool analysis, a median age assignment would yield an average error of 22.45 years; however, given differences in cohort size and composition, our WGS stool age predictions accuracies are consistent with the literature. As seen in other age prediction modeling, simpler models (e.g., RF) in prior studies often plateau at 7–10 years MAE for gut microbiomes[7]. With TRPCA, the deep residual network outperformed such baselines for our datasets, reducing prediction error by a significant margin. It is worth noting that microbiome clock performance varies by data type: 16S profiles typically yield lower accuracy than shotgun metagenomes, due to limited taxonomic resolution and the lack of fungal signals[67]. Consistent with this, our shotgun-based models outperformed 16S-based ones (as reflected in lower MAEs and higher $R^2$).

From a statistical standpoint, we took measures to ensure the rigor of our performance estimates. We used hold-out validation and cross-validation to confirm that our MAE improvements were not overfitting. Additionally, model comparisons were made to highlight performance improvements for the various age prediction tasks. TRPCA significantly outperformed all models for 5 out of 15 total regression tasks, with the KNN, SVR, and RF frequently performing on par with TRPCA (Fig. S23). Still, we acknowledge that the absolute $R^2$ values are modest, especially for 16S stool samples (Figs. 2c, 6a); however, these findings are consistent across model architecture and two independent datasets, implying that 16S sequencing methods current do not resolve the human microbiome enough for aging studies. While TRPCA showed significant numerical improvements across datasets (Fig. S23) with a 28% MAE reduction for skin WGS data (from $11.14 \pm 2.65$ to $8.03 \pm 4.10$), we acknowledge that overlapping uncertainty intervals in some cases indicate variable statistical significance of these improvements. These performance gains, though meaningful for computational modeling, likely offer limited additional clinical relevance compared to conventional ML approaches given our focus on chronological age in

https://doi.org/10.1038/s42003-025-08590-y **Article**

healthy individuals, rather than biological age between healthy and unhealthy cohorts. Nevertheless, the consistent performance pattern across body sites and sequencing methods reinforces TRPCA's value as a methodological advancement for microbiome-based age prediction research. Our sensitivity analysis of prediction residuals further validates the robustness of these findings. For many demographic variables, including sex and diet type, we observed no significant effects on age prediction residuals across most body sites and sequencing methods, suggesting our models are generally unbiased toward these factors. However, significant associations between residuals and certain variables were detected, particularly study-specific effects in 16S skin and oral samples and Body Mass Index(BMI) effects in WGS samples (Fig. S24, Supplemental Data 1–2). These findings highlight potential methodological confounders and physiological factors that could influence microbiome-based age predictions. A notable limitation is the incomplete metadata across datasets, with variables like non-Westernized status, diet, and BMI category frequently missing, preventing comprehensive assessment of all potential confounding factors across all body sites and sequencing methods. This metadata sparsity underscores the need for more complete demographic and lifestyle information in future microbiome aging studies to disentangle technical and biological influences on prediction accuracy. Given the results of the analysis, TRPCA is a promising tool for exploring the microbiome biological clock and has provided a substantial baseline for healthy aging. The analysis also highlights recommendations for methods and body sites which best support the investigation of the rate of aging.

Our study demonstrates that TRPCA significantly improves age prediction accuracy from microbiome profiles across various body sites and sequencing methods. We observed substantial reductions in MAE for age prediction, with improvements of 14% for skin (16S) and 28% for skin (WGS) compared to the best performing conventional ML methods (KNN for 16S and WGS skin samples, Table 1). These findings underscore the potential of transformer-based approaches in capturing complex patterns within microbiome data. The MTL approach of TRPCA further enhanced performance in simultaneous age prediction and birth country classification tasks, achieving a 6% reduction in MAE for age prediction and a 13% increase in accuracy for country classification. These results demonstrate the versatility of TRPCA in handling multiple related tasks efficiently.

Our integrated TRPCA and SHAP analysis revealed distinct age-related microbial signatures across body sites. In the skin, *Corynebacterium simulans* is more abundant in older individuals, while *Lactobacillus* species and *Cutibacterium acnes* are prevalent in younger skin, with *Staphylococcus epidermidis* increasing with age. In the oral cavity, younger individuals typically exhibit higher levels of *Actinomyces*, whereas *Fusobacterium* is enriched with age and *Veillonella* tends to decrease. For the gut, beneficial taxa such as *Akkermansia muciniphila* and *Faecalibacterium prausnitzii* are common in younger individuals; conversely, *Bacteroides* increase with chronological aging, and *Bifidobacterium* show reduced levels in older populations.

In conclusion, TRPCA represents an advancement in microbiome data analysis, offering improved accuracy in age prediction and valuable insights into age-associated microbial features. This approach paves the way for a more nuanced understanding of the relationship between the microbiome and chronological aging, potentially informing future interventions aimed at promoting healthy aging through microbiome modulation and the exploration of biological age from a microbiome perspective. Our findings support the application of transformer-based methods in microbiome research and suggest promising avenues for the adoption of state-of-the-art model architectures in microbiome analysis.

## Methods

### 16S rRNA microbiome data
16S data were downloaded directly from the Github repository of Huang et al.[20]. The samples include a total of 8959 samples from 10 studies (4434 fecal samples, 2550 saliva samples, and 1975 skin samples). Data were from

subjects with self-reported ages of 18 to 90 years, BMI of 18.5 to 30 kg/m², no reported inflammatory bowel disease or diabetes, and no antibiotic consumption 1 month before sampling. Pregnant, hospitalized, disabled, or critically ill individuals were excluded. For gut microbiota, the majority of acquired samples were derived from two projects: (i) the American Gut Project (AGP) and (ii) the Guangdong Gut Microbiome Project (GGMP). For oral and skin microbiota, samples were obtained from Qiita matching the inclusion and exclusion criteria above. Samples from the same host were grouped to ensure samples from the same host could not appear in both the test and train datasets, and to avoid training and predicting non-independent data.

### WGS microbiome data
WGS microbiome data were obtained from control and healthy samples in curatedMetagenomicData3[69]. Relative abundance of each feature was multiplied by sampling depth, then rounded to the closest whole number to create a count table for input into the TRPCA pipeline. The resulting dataset included a total of 9356 WGS samples, with 115 skin (Italy, $n = 48$; United States, $n = 27$; and others), 600 saliva (United States, $n = 451$; and others), and 8641 fecal (United Kingdom, $n = 2834$; the Netherlands, $n = 1,135$; United States, $n = 953$; and others) samples. Host age ranged from 0 to 91 years and BMI between 11.12 and 53.2 kg/m². All samples were labeled as 'healthy' and 'control' in the metadata categorical variables 'study_condition' and 'disease', respectively.

### Statistics and reproducibility
The analysis employs a combination of multivariate and permutation-based approaches to evaluate microbiome variation. PERMANOVA was used to quantify the effect sizes ($R^2$) of demographic and study variables on microbial community structure using Bray-Curtis dissimilarity matrices, which measure compositional differences between samples without considering abundance-weighted phylogenetic relationships[70,71]. For the sensitivity analysis of age prediction residuals, a permutation-based approach tested associations with categorical variables using Kruskal-Wallis tests and with continuous variables using Spearman rank correlations, generating empirical null distributions through 1000 random permutations to calculate p-values without parametric assumptions[72–74]. All resulting p-values were adjusted for multiple comparisons using the Benjamini-Hochberg false discovery rate (FDR) procedure to control for Type I errors while maintaining statistical power[75].

### RPCA
Robust Aitchison PCA is a dimensionality reduction technique where PCA is performed by log-transforming nonzero values before centering the data (RCLR), followed by a PCA dimensionality reduction[29]. Microbiome datasets were preprocessed, dropping features with low prevalence (present in less than 25 samples) and log-transformed via RCLR. Dimensionality reduction was completed via PCA (scikit-learn version: 1.3.0).

### TRPCA model and evaluation
The TRPCA model (Fig. 1) processes the RPCA vectors through a normalized transformer architecture with a multi-view projection for the regression and classification tasks. To begin, microbiome samples are transformed using the RPCA transform to generate the RPCA input embedding:

$$RPCA\ embedding = [PC_1, PC_2, ...PC_{n\ dimensions}]$$

The RPCA vectors are then projected into a higher-dimensional latent space using a linear layer, followed by L2 normalization. The high-dimensional latent vectors are then transformed into multiple views, which allows for the reshaping of the vectors into the shape [batch, PCA_dim, projection_dim]. The projection of PCA vectors synthetically created a projection_dim embedding for each principal component in the sample, while treating the PCA_dim as a sequence. The multi-view sequences are

then suitable for processing through a normalized transformer block with multi-head attention and MLP blocks with residual updates:

$$h_A = Norm\big(x + \alpha_A \cdot (Attention(x) - x)\big)$$

and

$$h_M = Norm\big(x + \alpha_M \cdot (MLP(x) - x)\big)$$

implementing learnable scaling parameters and L2 normalization for each update. The final outputs are aggregated using global average pooling across the projection dimensions before prediction using a linear layer for regression or MTL prediction.

To benchmark model performance for each regression task, we compared the MAE as a result of TRPCA to that obtained using SVR, GBR, KNN, NN, and RF (scikit-learn version: 1.3.0). We used 5-fold cross-validation to evaluate each regression algorithm. The following parameters were used: SVR with an RBF kernel ($C = 1.0$, epsilon $= 0.1$), Gradient Boosting Regression with 100 estimators (random state $= 42$), K-Neighbors Regression with 5 neighbors, Neural Network Regression with a Multi-layer Perceptron (hidden layer size $= 100$, ReLU activation, Adam solver, max_iter $= 1000$), and Random Forest Regression with 100 trees (random state $= 42$). Cross-validation scores and mean scores were computed for each regressor, ensuring a robust and comprehensive comparison of model performance in predicting age using microbiome data. This method was consistently applied to all datasets in the study. TRPCA models were trained using the model parameters (PCA_dim $= 256$, num_transformer_layers $= 1$, nhead $= 8$, projection_dim $= 4$, dropout $= 0.0$) for all models besides WGS skin where feature_size (number of principal components) was 64 due to the dataset sample size ($n = 115$) and the specifications of PCA dimensionality reduction requiring the number of dimension to be less than the number of samples. TRPCA models were trained with MSE Loss as the criterion for regression tasks and Cross Entropy for classification tasks. Additionally, Stochastic Gradient Descent (learning_rate $= 9e$-$3$) was used as the optimizer for MTL TRPCA models and AdamW for the TRPCA regression models. All model trainings used a Cosine Annealing with Warm Restarts scheduler. Batch size, learning rate, and the number of training epochs were modified if the model training demonstrated unstable convergence behavior. Multiple samples from the same subject were grouped using GroupKFold (scikit-learn version: 1.3.0) for all models to avoid biasing model predictions to any unique individual. StratifiedKFold, StratifiedGroupKFold and KFold were also used for CVs for regressions and classifications which required stratifying train/test sets or CVs that required no grouping or stratification (CV $= 10$ for all tasks). For CV training experiments and exploration, a single A100 GPU(80GB RAM) was used for training via Google Colab, allowing for training times of 1–15 min depending on the size of the dataset and model size experimentation. L4/T4 GPUs may also be used for training with minimal decrease in training time.

## SHAP values for interpretable feature extraction

To investigate the mechanisms underlying the predictions, we used an explainable AI algorithm called SHAP. SHAP is advantageous because it can work with any machine learning model, including tree-based models such as RF, as well as transformer models. SHAP combines game theory with local explanation, enabling accurate interpretations of how the model predicts a particular value for a given sample. These explanations, called local explanations, show how each feature contributes, either positively or negatively, to the prediction of a specific instance. SHAP provides a ranked list of important features for a machine learning model and additionally explains how each impactful feature contributes to the prediction of specific values. This local explanation approach reveals subtle changes and interrelations that might be missed when differences are averaged out. SHAP offers a global view of local explanations through the Shapley value matrix, which consists of one row per data instance and one column per feature. The Shapley values can be visualized as summary dot or bar plots, which help

interpret the entire model. These plots show the impact of each feature on the prediction of the target variable for instances with similar feature values[76,77]. The SHAP (version 0.45.0) values derived from the model are indicative of the importances between principal components from the RPCA transformation. Therefore, in order to trace back the importances on the feature level, we took the dot product of the SHAP value matrix (n samples by j principal components) and the PCA feature loading matrix (j principal components by k microbiome features). The dot product of the SHAP matrix and PCA feature loading matrix yields a weighted feature importance matrix with the original features for each sample.

$$\text{FeatureImportance}_{(nsamples,\ kfeatures)}$$
$$= \text{SHAPMatrix}_{(nsamples,\ jcomponents)}\text{PCAComponents}_{(jcomponents,\ kfeatures)}$$

## Multi-task learning (MTL)

Multi-task learning is a powerful technique in machine learning that trains a model on multiple related tasks simultaneously to improve model performance and save compute time and resources[78]. MTL can be applied to supervised and unsupervised learning, including differing objectives such as a combination of regression and classification tasks. To create an implementation of MTL for the TRPCA model, the output from the model's transformer encoder was passed to both a classification head and an additional attention layer for the regression task. The regression attention and classification outputs were then passed to the regression head for host age prediction.

## Reporting summary

Further information on research design is available in the Nature Portfolio Reporting Summary linked to this article.

## Data availability

Datasets for 16S skin, oral and gut samples can be found in the Github repository https://github.com/shihuang047/age-prediction. WGS datasets can be retrieved using the code provided at https://waldronlab.io/curatedMetagenomicDataAnalyses/articles/MLdatasets.html and using only the healthy/control subject dataset. Paired data from THDMI are part of Qiita study 10317 and European Bioinformatics Institute accession number PRJEB11419. Paired samples from the FINRISK dataset are protected with access details available in the European Genome-Phenome Archive under accession number EGAD00001007035. The source data behind the graphs in the paper can be found in file named Supplementary Data 1-3.

## Code availability

Analysis notebooks available at https://github.com/tydymy/TRPCA_analysis and TRPCA model repository can be found at https://github.com/tydymy/TRPCA[79].

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

## Acknowledgements

This work was funded by Danone Nutricia Research and the Center for Microbiome Innovation and supported by the Microsetta initiative. B.D.P. was supported by the Research Foundation Flanders (grant numbers 1S04122N and V477223N).

## Author contributions

T.M., S.J.S. and A.B.: Conducted, analysed and interpreted experiments. T.M., S.J.S., S.K., R.K. and A.B.: Conceived, designed and/or supervised parts of the study. T.M., S.J.S., Y.C., B.D.P., L.K., D.M., S.H., A.H., L.L., G.R., M.L., S.A.S., C.C., R.G., S.K., R.K. and A.B.: Contributed to the writing of the paper.

## Competing interests

G.R., M.L., and S.A.S. are Employees of Danone. D.M. is a consultant for BiomeSense, Inc., has equity and receives income. The terms of these arrangements have been reviewed and approved by the University of California, San Diego in accordance with its conflict of interest policies. A.B. is a founder of Guilden Corporation and is an equity owner. The terms of these arrangements have been reviewed and approved by the University of California, San Diego in accordance with its conflict of interest policies. R.K. is a scientific advisory board member, and consultant for BiomeSense, Inc., has equity and receives income. He is a scientific advisory board member and has equity in GenCirq. He is a consultant and scientific advisory board member for DayTwo and receives income. He has equity in and acts as a consultant for Cybele. He is a co-founder of Biota, Inc., and has equity. He is a co-founder of Micronoma and has equity and is a scientific advisory board member. The terms of this arrangement have been reviewed and approved by the University of California, San Diego, in accordance with its conflict of interest policies. All other authors declare no competing interests.

## Additional information

[1]Center for Microbiome Innovation, Jacobs School of Engineering, University of California San Diego, La Jolla, CA, USA. [2]Shu Chien-Gene Lay Department of Bioengineering, University of California San Diego, La Jolla, CA, USA. [3]Department of Dermatology, University of California San Diego, La Jolla, CA, USA. [4]Department of Pediatrics, University of California San Diego, La Jolla, CA, USA. [5]Biomedical Sciences Graduate Program, University of California San Diego, La Jolla, CA, USA. [6]Center for Microbial Ecology and Technology, Ghent University, Ghent, Belgium. [7]Neuroscience Graduate Program, University of California San Diego, La Jolla, CA, USA. [8]Faculty of Dentistry, The University of Hong Kong, Hong Kong SAR, China. [9]Finnish Institute for Health and Welfare, Helsinki, Finland. [10]Institute for Molecular Medicine Finland, FIMM-HiLIFE, Helsinki, Finland. [11]Department of Computing, University of Turku, Turku, Finland. [12]Danone Research and Innovation, Utrecht, the Netherlands. [13]Bioinformatics and Medical Informatics Program, San Diego State University, San Diego, CA, USA. [14]Department of Biology, San Diego State University, San Diego, CA, USA. [15]Department of Computer Science and Engineering, University of California San Diego, La Jolla, CA, USA. [16]Halıcıoğlu Data Science Institute, University of California San Diego, La Jolla, CA, USA. ✉e-mail: abartko@ucsd.edu

