## [Transparent Peer Review file · Communications Biology]

Chronological Age Estimation from Human Microbiomes with Transformer-based Robust Principal Component Analysis

Corresponding Author: Dr Andrew Bartko

Version 0:

Reviewer comments:

Reviewer #1

(Remarks to the Author)

Summary:

Myers et al. introduced Transformer-based Robust Principal Component Analysis (TRPCA), integrating transformer architectures with RPCA for microbiome analysis. Using 8,959 16S and 9,356 WGS samples, TRPCA outperformed conventional machine learning models in age prediction across body sites, reducing MAE by 14-28% for skin microbiomes. In a multi-task learning framework, it also predicted the host birth country from WGS data. SHAP analysis identified key age-associated bacterial species. Paired sample analysis showed a strong correlation ($R^2=0.63$) in age prediction error residuals between 16S and WGS despite differing predictive performances. While the study has demonstrated the potential of using TRPCA in microbiome-based age prediction, some areas could be further clarified and improved.

Major comments:

1. Please evaluate the statistical and practical significance of the performance improvement of TRPCA over other models. For skin WGS data, the MAE decreased by 28% (from 11.14 ± 2.65 to 8.03 ± 4.10), yet the uncertainty intervals overlap, casting doubt on the significance of this difference. Additionally, discuss whether such an MAE reduction is biologically or clinically meaningful in the context of age prediction.
2. Assess the confounding effects of BMI on the prediction model, noting the distinct BMI ranges in the 16S (18.5 - 30.0) and WGS (11.12 - 53.20) datasets. Please also conduct a sensitivity analysis to evaluate the effects of other potential confounders, such as diet, medication use, or geography, on model performance.
3. Please explain the moderate correlation observed between paired WGS oral and stool samples. Elaborate on the biological and technical factors underlying this correlation pattern and relate these findings to the existing literature on body site-specific microbiome variations.
4. Explore in greater detail the high correlation in age prediction residuals between 16S and WGS methods. Consider whether this pattern indicates systematic biases or important biological factors not captured by either method. Discuss how understanding these residuals could inform the development of improved age prediction models and provide insights into method-independent aging signatures.
5. Discuss the biological implications of the findings beyond predictive accuracy. For instance, how do the identified age-associated taxa relate to aging processes or health outcomes?

Minor comments:

1. Increase the font size in all figures to improve readability. The text, labels, and legends are currently too small to be comfortable reading.
2. Line 290: The statement "the exploration of model parameters, learning rates, and weighted loss functions beyond TRPCA's defaults could further improve model performance" is not appropriate because one can also tune the hyperparameters of other models (e.g., RF, NN) to improve the performance of these "competitors."
3. Line 334: Ten-fold cross-validation was applied to the paired sample analysis, while five-fold was used for the main

analysis. Provide a rationale for this difference.

4. Line 461: The discussion of computational overhead lacks specificity. Please include more details, such as the time required to train TRPCA once with a given set of hyperparameters, and specify the hardware used. This will enable readers to evaluate the method's computational feasibility.

5. Figure 4 appears before Figure 3, which disrupts the manuscript's logical flow.

6. Figure 4 could be more informative if the confusion matrix of RF is placed side-by-side with the current matrix for MTL.

Reviewer #2

(Remarks to the Author)

I read the paper with interest and believe its findings, beyond their computational novelty, could offer biological and clinical insights into aging-related dysbiosis tied to age prediction. Below are my comments, which I hope will further enhance the reader's experience if applied.

The primary goal of the paper titled "Transformer-based Robust Principal Component Analysis Improves Chronological Age Prediction from Human Microbiomes" is to introduce and evaluate a novel deep learning approach, Transformer-based Robust Principal Component Analysis (TRPCA), for predicting chronological age from human microbiome data. The authors aim to demonstrate that TRPCA outperforms conventional machine learning methods in this task while providing interpretable insights into microbial features associated with aging. However, the paper lacks substantial review of previous research in deep learning based age prediction, such as description of the first deep learning based microbiome aging clock by Galkin et al (2018 on preprint, 2020 in Cell). The publication would benefit from clearer implications for the aging process.

1. Title. I would suggest making the title less deterministic by avoiding the word "improves". Improves compared to what? Also, please define whether the most important aspect of your paper is the Transformer-based Robust Principal Component Analysis or Chronological Age prediction. I believe highlighting the age prediction would be beneficial due to potential application e.g in clinical settings. I would suggest "Chronological Age Estimation from Human Microbiome with Transformer-based Robust Principal Component Analysis" or "Transformer-based Robust Principal Component Analysis as a promising tool in chronological age estimation from human microbiome samples"

2. The paragraph from lines 56-63 needs more references and explanation. The authors overuse the different versions of "has impact" or "is implicated" without providing any indication of a nature of such implication. Moreover, the introduction should specify which properties of the microbiome have such "implication": taxonomy, microbial load?

2a. Whenever the authors refer to impact on aging as in line 58, the concept should be narrowed down so that the literature supports it. I usually recommend using the rate of aging, if applicable.

2b. The statement in line 63 "The gut, skin and oral microbiomes have all been implicated in aging" should have been supported by at least one reference for each, as well as briefly explain what kind of implication the research provides.

3. The introduction would benefit from stating that dysbiosis (changes to the microbiome) was recently added as a new hallmark of aging. <https://pubmed.ncbi.nlm.nih.gov/36599349/>

4. In line 65, alpha and beta diversity should be briefly defined to ensure clarity. Moreover, as per my previous comments, the sentence "linked to chronological and biological aging." should be expanded. Was the property linked with accelerated aging?

5. The paper would benefit from a brief introduction of the difference between chronological and biological aging.

6. Sentence in lines 73 and 74 "As individuals age, changes in the oral microbiota 74 mirror the physiological changes occurring in the host." should be supported by a reference.

7. As per my previous comments, the age-related changes to skin microbiome should be more detailed (taxonomy, load, increased, decreased, etc?)

8. Sentence in line 86 "Differential abundance analysis (...) should be supported by a reference.

9. Line 88 "(as measured by corneometer)" seems redundant

10. In line 100, "(RF, SVM)" should use full names, and state that they are ML methods

11. In line 116 "Transformer models (...)" should include a reference, or if there is not one, should state potential directions of research, while highlighting that this kind of research has not been published yet

12. Please double check your statement in line 128 "improves age prediction by >10% for all skin (...)" because in table 1 you have some lower improvements (e.g., 2% for 16S gut)

13. In line 163, RPCA drops "features with low prevalence" before RCLR, but the threshold for "low prevalence" is unspecified. Please specify for reproducibility

14. The references don't follow a standardized way and are not formatted to journal guidelines (see at <https://www.nature.com/commsbio/submit/submission-guidelines#references>) Reference number 18 is not properly cited according to journal standards. Reference 36 is not properly cited. Moreover, since 36 is a preprint, it should either be replaced by a peer-reviewed article, or it should be noted in the manuscript that this reference was not supported by peer review. 58, 51, 47, 44 are also not properly referenced.
15. The authors reference two models for DL for microbiome-derived disease prediction, however, it is not relevant to the scope of their research (age prediction). It is imperative that previous work in the field is appropriately referenced.
- 15a. The authors did not reference the seminal paper for DL for age prediction [https://www.cell.com/iscience/fulltext/S2589-0042\(20\)30384-9](https://www.cell.com/iscience/fulltext/S2589-0042(20)30384-9) which was the first research attempt using the microbiome data and DL for age prediction. Moreover, this model is patented with a US patent <https://patentimages.storage.googleapis.com/31/dc/2e/be5f9918f01378/WO2020084536A1.pdf>
- 15b. The paper would benefit from referencing this research as an example of previous work <https://academic.oup.com/bib/article/22/3/bbaa073/5835556#248048567>
- 15c. After introducing background evidence as mentioned above, the authors should briefly differentiate between the types of DL used above, and their model: how are they different (in a language that a physician/ not computational biologist would understand)
16. In line 256, the sentence references a comparison to "other methods" but Figure 2 doesn't utilize them. In this case, studies with the "other methods" should be referenced, and the sentence would benefit the Discussion section, rather than results
17. Also in line 256, I would refrain from using the word "significant" unless you are referring to statistical significance
18. The MAE improvement percentages should be reported with standard deviations to show consistency across folds, if available
19. Here is a tiny and petty detail, please remove the space in front of the fullstop in line 285:)
20. In lines 310-311 "E. faecalis was the most important feature for age prediction" please specify whether it was just the presence or abundance or other of this bacterium.
21. In line 321 "For example, for the age group 28-35, (...)" I believe this sentence is one of the clinical pearls of this research, and the features driving the age to older/younger should be discussed in more detail. I believe this finding is so good and so industry relevant that it requires description in detail and its own subsection. So far all the relevant information gets lost in the technical description of the model, and readers are likely to miss it, which would be a shame
22. I believe the paragraph about transformers lines 352-363 belongs in the introduction. If I was a non computationally trained reader, I would have largely benefited from this description before reading the results. In the first sentences of "discussion", however, I expect a clear and brief description of what was done and what it means, not the technical details. In reality, your discussion starts in line 365.
- 22a. Speaking of discussion, please highlight the finding of "divers" of skewed age prediction. I humbly believe this is one of your most important findings and definitely needs more attention
23. Line 378, as per my previous comments, what kind of correlation? Does the abundance of this bacterium drive or deaccelerate the rate of aging?
24. I believe your discussion of bacteria types and your findings should be restructured like "description of bacterium's one (specific!) implication in aging. Your model's findings about bacterium 1." then move to bacterium 2 and your findings about bacterium 2. In this way, you will avoid the feeling that it is introduction again.
25. For your discussion of gut microbiome (415-431) there is not a single reference to your findings. Please, change that.
26. Line 431, please modify "improves age prediction" to "improves age prediction accuracy" and state the model of comparison that indicates this improvement, and by how much
Same for 6% reduction in line 440: compared to what?
27. I believe the paragraph in lines 444-449 needs more attention to detail. Specifically, please differentiate between which bacteria are the drivers and which are the deaccelerators of the rate of aging, and whether you refer to their general abundance, detention, or other parameters.
28. Limitations belong in discussion section
29. The publication would benefit from a brief sentence about the implications of the research. Is it an industry device

potential? Clinical settings? How can it be useful outside of the lab?

30. The discussion lacks MAE comparison to other papers using DL to measure age such as aforementioned [https://www.cell.com/iscience/fulltext/S2589-0042\(20\)30384-9](https://www.cell.com/iscience/fulltext/S2589-0042(20)30384-9) with MAE=5.9 based on microbiome taxonomic profiles (the comparison should be more detailed)

31. Another limitation of the paper is the lack of biological age prediction, a more clinically relevant metric than chronological age, which could be noted as "further research"

32. The statistical analysis lacks rigor, omitting significance testing (e.g., t-tests) for MAE reductions—a practice increasingly expected despite being optional in some ML papers—while failing to address low R^2 values (e.g., 0.17 for 16S gut) that suggest poor fit and high variance in WGS skin (± 4.10 , $n=115$) that questions reliability. I suggest adding statistical tests, discussing R^2 implications, and exploring variance drivers (e.g., small sample size) to align with best practices and bolster claims.

33. I believe you have a typo in table 1 Skin (16S) stating your MAE as 5.09 ± 1.09 , but in Figure 2 you state it as 5.09 ± 1.07 .

Version 1:

Reviewer comments:

Reviewer #1

(Remarks to the Author)

I have no further questions

Reviewer #2

(Remarks to the Author)

Great job on improving the manuscript!

Response to Reviewers:

We appreciate the reviewer's continued engagement with our work and their thoughtful comments. Below are our responses to each of the concerns. We have included excerpts of the changed manuscript text for ease of review. These excerpts are in quoted text in our response to the reviewer comments.

Reviewer #1:

Comment #1:

Please evaluate the statistical and practical significance of the performance improvement of TRPCA over other models. For skin WGS data, the MAE decreased by 28% (from 11.14 ± 2.65 to 8.03 ± 4.10), yet the uncertainty intervals overlap, casting doubt on the significance of this difference. Additionally, discuss whether such an MAE reduction is biologically or clinically meaningful in the context of age prediction.

Response #1:

To evaluate the statistical and practical significance of the model performance over other models, we conducted a t-test comparison of TRPCA versus the other models. We found that, for many of the comparisons, TRPCA demonstrated a significant improvement over other model architectures. TRPCA significantly outperformed all models for 5 out of 15 total regression tasks, with the KNN and RF frequently performing on par with TRPCA. The results from this analysis are summarized in **Figure S23** for the regression tasks. Additionally, the significance of the improvements are discussed here (lines 631-637):

“While TRPCA showed significant numerical improvements across datasets (**Figure S23**) with a 28% MAE reduction for skin WGS data (from 11.14 ± 2.65 to 8.03 ± 4.10), we acknowledge that overlapping uncertainty intervals in some cases indicate variable statistical significance of these improvements. These performance gains, though meaningful for computational modeling, only offer incremental clinical relevance compared to conventional ML approaches because of our focus on chronological age in healthy individuals in this study. “

Comment #2:

Assess the confounding effects of BMI on the prediction model, noting the distinct BMI ranges in the 16S (18.5 - 30.0) and WGS (11.12 - 53.20) datasets. Please also conduct a sensitivity analysis to evaluate the effects of other potential confounders, such as diet, medication use, or geography, on model performance.

Response #2:

The models were stratified by country, study, and age bin for training, given complete metadata for these fields. We acknowledge the potential confounders like diet, BMI, and medication use on host age prediction, however, we have varying completeness of these metadata due to the multi-study aspect of the analysis. To address the confounding variables, we conducted a post-hoc permutation analysis on the age prediction residuals, comparing categorical variables using Kruskal-Wallis tests and continuous variables using Spearman rank correlations. All resulting p-

values were adjusted for multiple comparisons using the Benjamini-Hochberg false discovery rate. The summary of the significant variables by body_site and data type are included in **Table S4**, and a more detailed description in **Table S3**. Lastly, the effect size of variables were analyzed using PERMANOVA on Bray-Curtis dissimilarities in **Figure S24**. The limitations of the data availability and the results of the residuals analysis are outlined in lines 624-656, with specific discussion of the potential confounders here:

“Our sensitivity analysis of prediction residuals further validates the robustness of these findings. For many demographic variables, including sex and diet type, we observed no significant effects on age prediction residuals across most body sites and sequencing methods, suggesting our models are generally unbiased toward these factors. However, significant associations between residuals and certain variables were detected, particularly study-specific effects in 16S skin and oral samples and BMI effects in WGS samples (**Figure S24, Table S2-3**). These findings highlight potential methodological confounders and physiological factors that could influence microbiome-based age predictions. A notable limitation is the incomplete metadata across datasets, with variables like: non-Westernized status, diet, and BMI category were frequently missing, preventing comprehensive assessment of all potential confounding factors across all body sites and sequencing methods.”

Comment #3:

Please explain the moderate correlation observed between paired WGS oral and stool samples. Elaborate on the biological and technical factors underlying this correlation pattern and relate these findings to the existing literature on body site-specific microbiome variations.

Response #3:

We appreciate the attention given to the host specific residual correlation. To address the factors associated with the residual correlations, we have added the Biological and Technical Insights from Residual Age Predictions (Line 540) section to the text. Lines 541-553 discuss the WGS oral and gut residuals explicitly, explaining how the moderate correlation strength could be due to the oral microbiome’s reported advantage in predicting chronological age, compared to stool microbiome samples. Additional discussion is included pertaining to body site specific characteristics which may be involved in dampening the correlation. Discussion for the paired samples is outlined below, with references to “The oral-gut microbiota relationship in healthy humans: identifying shared bacteria between environments and age groups”, “Human skin, oral, and gut microbiomes predict chronological age”, and “Phylogeny-Aware Analysis of Metagenome Community Ecology Based on Matched Reference Genomes while Bypassing Taxonomy” to relate our findings to the literature:

“Paired whole-genome shotgun (WGS) microbiome samples from different body sites showed only a moderate alignment in predicted aging. Specifically, we observed a modest correlation ($R^2 \approx 0.34$) between age predictions from oral versus stool WGS microbiomes. This level of concordance likely reflects fundamental microbiome body-site differences. The oral and gut communities are compositionally distinct, with over 90% of taxa in each habitat are unique and not shared with the other, limiting cross-site prediction agreement⁷⁷. Prior studies confirm that

microbiome aging signatures are highly site-specific: for example, the skin microbiome can predict age much more accurately than the gut microbiome, potentially obfuscating the correlation strength between the paired residuals²¹. Thus, an individual's oral and gut microbiotas may each track aspects of host aging, but differences in local environment, diet, and host factors at each site dampen the correlation. Nonetheless, the moderate correlation underscores that chronological age imprints on each body-site microbiome in a distinct manner, in line with extensive evidence that the microbiome is compartmentalized⁷⁷.

In contrast, we found a much stronger residual correlation ($R^2 \approx 0.63$) between 16S rRNA gene versus WGS age predictions for the paired stool samples. This high correspondence indicates that when one sequencing modality's model over- or under-predicted a person's age, the other tended to show a similar residual. Several factors may explain this alignment of residuals. First, it could reflect shared biological signals captured by both methods: the individuals who are hard to predict (or biologically unusual for their age) via 16S are likewise outliers via WGS. In other words, method-independent aging features drive the prediction errors in both cases. This is a promising scenario, suggesting our models are detecting real host-microbiome age relationships rather than random noise."

Comment #4:

Explore in greater detail the high correlation in age prediction residuals between 16S and WGS methods. Consider whether this pattern indicates systematic biases or important biological factors not captured by either method. Discuss how understanding these residuals could inform the development of improved age prediction models and provide insights into method-independent aging signatures.

Response #4:

In line with the expansion upon the paired WGS stool and oral sample residuals, we expanded on the 16S and WGS methods as well (Lines 554-577). In this text, we review implications of systematic biases, as well as potential biological factors driving the age prediction models. We greatly appreciate this feedback, as the expanded discussion highlights aspects of chronological aging which have yet to be fully explored. Below is the updated discussion of the paired 16S and WGS stool residual correlation:

"In contrast, we found a much stronger residual correlation ($R^2 \approx 0.63$) between 16S rRNA gene versus WGS age predictions for the paired stool samples. This high correspondence indicates that when one sequencing modality's model over- or under-predicted a person's age, the other tended to show a similar residual. Several factors may explain this alignment of residuals. First, it could reflect shared biological signals captured by both methods: the individuals who are hard to predict (or biologically unusual for their age) via 16S are likewise outliers via WGS. In other words, method-independent aging features drive the prediction errors in both cases. This is a promising finding, suggesting our models are detecting real host-microbiome age relationships rather than random noise. Alternatively, systematic biases in sample processing or subject selection might influence both 16S and WGS similarly. For instance, if certain subpopulations cause microbiome age to deviate from chronological age, both sequencing methods could

identify these deviations. We also consider the role of taxonomic resolution with shotgun sequencing providing finer species level detail absent in 16S profiling. In theory this should improve accuracy, and indeed shotgun models often modestly outperform 16S⁷⁸. However, the high residual correlation implies that, although the age prediction accuracy with WGS data was lower due to improved species resolution, the associated deviation in model prediction remains consistent, posing the age prediction residual as a reproducible measure. Such method-independent agreement supports the idea that the residuals reflect true biological variation in aging pace, as opposed to method-specific artifacts. In summary, while different sequencing technologies have inherent biases, the convergence of their age predictions in our study suggests we have identified robust aging signatures in the gut microbiome that transcend technical platforms. This is encouraging for future efforts to develop microbiome-based age biomarkers, as it indicates a person's microbiome age is not merely an artifact of the sequencing method but rather grounded in biological signals.”

Comment #5:

Discuss the biological implications of the findings beyond predictive accuracy. For instance, how do the identified age-associated taxa relate to aging processes or health outcomes?

Response #5:

We appreciate the implications of age-associated taxa in the context of the aging process and health outcomes. To expand upon the identified taxa, we added Lines 578-585 to highlight the feature extraction method's ability to isolate the microbiome drivers on a per sample level, while pulling out relevant taxa associated with the aging process. Given the study's focus on healthy individuals, addressing specific health outcomes and biological aging is currently a limitation. However, the ability to address specific health outcomes requires higher age prediction accuracy than what has been possible previously. The advancements described in this manuscript may enable greater predictive power for those health outcomes. We appreciate the feedback provided and plan to build upon this work to address health outcomes and biological aging as a future direction. Features associated with aging are introduced with a skin microbiome example in lines 578-585:

“Our analysis of model feature importance by age group revealed that certain microbial features disproportionately influenced model predictions, notably in individuals aged 28–35 for WGS skin data (**Figure 5a**). In this younger adult cohort, the model tended to predict lower ages as a function of a feature's abundance. SHAP values highlighted two organisms in particular, *Cutibacterium granulorum* and *Malassezia globosa*, as drivers of skewed age estimates. By examining SHAP values, we identified concrete microbial candidates that may serve as sentinels of accelerated or decelerated microbiome aging in an individual based on the influence of the feature to drive model prediction toward younger or older ages.”

And then expanded upon for all body sites in lines 586-605:

“Beyond these two taxa, our models identified a broader panel of age-associated microbes across skin, gut, and oral sites, each with potential impacts on host health. For instance, a relative abundance of certain oral anaerobes (e.g. *Fusobacterium*) emerged as markers of an older oral microbiome in our models, consistent with literature showing periodontal bacteria increase with age in some individuals⁶. These organisms are well-known contributors to gum disease, which tends to worsen with age and can have systemic inflammatory effects. Their prominence in age-prediction models suggests that oral health status is reflected in the microbiome age signal. Likewise, in the gut we noted that model coefficients and SHAP values often flagged depletion of beneficial genera (e.g. *Faecalibacterium*) and enrichment of pro-inflammatory taxa (e.g. *Escherichia/Shigella*) in individuals predicted to be older than their chronological years. The ability to predict host age from microbiome data carries several translational implications. In the realm of preventive and personalized medicine, a microbiome age model could be developed into a biological age indicator alongside other metrics (epigenetic, proteomic, etc.). There are also clear opportunities in the skincare and cosmetic industry. The skin microbiome was a strong component of our models, and prior work shows it to be a reliable mirror of chronological age^{19,21}. Probiotic supplementation is another direction as our findings identify specific taxa (*Faecalibacterium* in the gut, or *Veillonella* in the oral cavity) that decline with age⁶. Restoring fiber-degrading, butyrate-producing bacteria in an elderly gut microbiome might not only make the microbiome profile younger but also confer benefits like improved colonic health, reduced inflammation, and better metabolic regulation⁷⁹. “

Comment #6:

Increase the font size in all figures to improve readability. The text, labels, and legends are currently too small to be comfortable reading.

Response #6:

To improve the readability of the manuscript, we increased the font size on all the main figures in the manuscript.

Comment #7:

Line 290: The statement "the exploration of model parameters, learning rates, and weighted loss functions beyond TRPCA's defaults could further improve model performance" is not appropriate because one can also tune the hyperparameters of other models (e.g., RF, NN) to improve the performance of these "competitors."

Response #7:

As this is a valid point and applicable to all of the model architectures, we removed the statement.

Comment #8:

Line 334: Ten-fold cross-validation was applied to the paired sample analysis, while five-fold was used for the main analysis. Provide a rationale for this difference.

Response #8:

For the TRPCA analysis, a CV of 10 was used for comparing the models; however, this was not explicitly stated in the methods section of the manuscript. The methods have been updated to reflect the cross validation of 10 folds. See line 335:

“StratifiedKFold, StratifiedGroupKFold and KFold were also used for CVs for regressions which required stratifying train/test sets or CVs that required no grouping or stratification (CV=10 for all tasks).”

Comment #9:

Line 461: The discussion of computational overhead lacks specificity. Please include more details, such as the time required to train TRPCA once with a given set of hyperparameters, and specify the hardware used. This will enable readers to evaluate the method's computational feasibility.

Response #9:

Thank you for your acknowledgement of the feasibility of training TRPCA models. We have added lines 337-341 in the methods section to describe the hardware, environment, and training times for TRPCA models on accessible hardware. See below:

“For CV training experiments and exploration, a single A100 GPU(80GB RAM) was used for training via Google Colab, allowing for training times of 1-15 minutes depending on the size of the dataset and model size experimentation. L4/T4 GPUs may also be used for training with minimal decrease in training time.”

Comment #10:

Figure 4 appears before Figure 3, which disrupts the manuscript's logical flow.

Response #10:

Noting the misordered figures, we have fixed the logical mentioning of figures in the text by reordering the figures.

Comment #11:

Figure 4 could be more informative if the confusion matrix of RF is placed side-by-side with the current matrix for MTL.

Response #11:

Figure 3 (formerly Figure 4) has been updated to include both the confusion matrix and regression plot from the two RF models to provide a comparison to the TRPCA MTL model. The figure now better demonstrates the model performances in a more informative manner, highlighting countries where TRPCA outperforms RF and the greater R2 for the TRPCA model, although the variance for the TRPCA model is greater than the RF model.

Reviewer #2:

Comment #1:

Title. I would suggest making the title less deterministic by avoiding the word "improves". Improves compared to what? Also, please define whether the most important aspect of your paper is the Transformer-based Robust Principal Component Analysis or Chronological Age prediction. I believe highlighting the age prediction would be beneficial due to potential application e.g in clinical settings. I would suggest "Chronological Age Estimation from Human Microbiome with Transformer-based Robust Principal Component Analysis" or "Transformer-based Robust Principal Component Analysis as a promising tool in chronological age estimation from human microbiome samples"

Response #1:

Based on your feedback about the significant findings and our own interests derived from the analysis, we would like to highlight the insights derived from the exploration of the model. We opted to change the title to "*Chronological Age Estimation from Human Microbiome with Transformer-based Robust Principal Component Analysis*", as the novel method highlights interesting aspects of age prediction and host associated age prediction residuals.

Comment #2:

The paragraph from lines 56-63 needs more references and explanation. The authors overuse the different versions of "has impact" or "is implicated" without providing any indication of a nature of such implication. Moreover, the introduction should specify which properties of the microbiome have such "implication": taxonomy, microbial load?

Response #2:

To address the concerns about the overuse of uninformative language, the introduction (lines 54-125) was rewritten to highlight specific taxa which change with age, highlighting many of the taxa we discuss in the results section. Some examples include:

“The human microbiome undergoes notable taxonomic and functional shifts as we age, influencing immune regulation, metabolism, dysbiosis, and disease risk^{1,2}. In general, aging is accompanied by the loss of certain beneficial microbes and a rise in opportunistic ones¹. For example, older adults tend to show reduced abundance of *Faecalibacterium prausnitzii* and related butyrate-producing gut bacteria, alongside expansion of pro-inflammatory taxa¹. On aged skin, sebaceous genera like *Cutibacterium acnes* decline, while *Corynebacterium* becomes more prominent^{3,4}. In the oral cavity, age is associated with a drop in commensals such as *Streptococcus* and *Veillonella*, and an enrichment of anaerobes like *Fusobacterium*, *Treponema*, and *Porphyromonas* that are linked to periodontal disease^{5,6}. Notably, these microbiome alterations correlate more with biological aging and frailty than with chronological age alone⁴, indicating that a healthy aging microbiome is characterized by specific taxonomic patterns rather than just overall diversity or load.”

“ Conversely, certain bacteria often considered beneficial for metabolic health, like *Akkermansia muciniphila*, actually rise in abundance in many healthy seniors¹. Exceptionally long-lived

individuals tend to harbor higher gut microbial diversity and are enriched in taxa such as *Lactobacillus*, *Akkermansia*, *Methanobrevibacter*, and butyrate-producing *Clostridiales*, which may contribute to their reduced inflammatory status and improved metabolic profiles¹². “

“For example, higher relative abundances of *Fusobacterium*, *Treponema*, and *Porphyromonas* are observed in older individuals, reflecting a microbiota more prone to gum disease and chronic inflammation⁵. This age-related oral dysbiosis can exacerbate periodontal disease, tooth loss, and even contribute to systemic inflammation, as oral pathogens have been linked to conditions like atherosclerosis and aspiration pneumonia.”

Comment #3:

Whenever the authors refer to impact on aging as in line 58, the concept should be narrowed down so that the literature supports it. I usually recommend using the rate of aging, if applicable.

Response #3:

We appreciate the specificity of using the correct language for describing aspects of the aging process. We updated the language and updated the text to specify "the rate of aging" (lines 535) or chronological aging (lines 63-73), where applicable.

“Positive correlations between *Bacteroides* and age have been reported, suggesting a potential role in the rate of aging^{71,72}.”

“Notably, these microbiome alterations correlate more with biological aging and frailty than with chronological age alone⁴, indicating that a healthy aging microbiome is characterized by specific taxonomic patterns rather than just overall diversity or load. Chronological age simply counts years lived, whereas biological age, estimated from integrative molecular and clinical biomarkers, more accurately reflects functional decline and health risk⁷. Emerging microbiome clocks operationalize this concept by training machine-learning models on gut, skin, and oral community profiles to infer host age; the predicted value, often termed microbiome age, can deviate markedly from chronological age⁸. The magnitude and direction of this deviation correlate with phenotypes of healthy aging: older microbiome age ($\Delta\text{Age} > 0$) aligns with frailty and increased four-year mortality, whereas a younger-than-expected microbiome is linked to enhanced metabolic and inflammatory resilience^{4,9-11}. “

Comment #4:

The statement in line 63 "The gut, skin and oral microbiomes have all been implicated in aging" should have been supported by at least one reference for each, as well as briefly explain what kind of implication the research provides.

Response #4:

We appreciate the reviewer's critique and have revised the manuscript to address this concern. The background section now includes examples of taxa, body site, and the directionality of the abundance shift to better frame the phenomenon of the rate of aging from the microbiome

perspective. A few examples of the updated implications can be found in lines 55-65, 74-79, and 93-96.

“In general, aging is accompanied by the loss of certain beneficial microbes and a rise in opportunistic ones¹. For example, older adults tend to show reduced abundance of *Faecalibacterium prausnitzii* and related butyrate-producing gut bacteria, alongside expansion of pro-inflammatory taxa¹. On aged skin, sebaceous genera like *Cutibacterium acnes* decline, while *Corynebacterium* becomes more prominent^{3,4}. In the oral cavity, age is associated with a drop in commensals such as *Streptococcus* and *Veillonella*, and an enrichment of anaerobes like *Fusobacterium*, *Treponema*, and *Porphyromonas* that are linked to periodontal disease^{5,6}. Notably, these microbiome alterations correlate more with biological aging and frailty than with chronological age alone⁴, indicating that a healthy aging microbiome is characterized by specific taxonomic patterns rather than just overall diversity or load.”

“In the gut, aging is typically marked by a shift away from a youth-associated community towards a dysbiotic profile. Studies in older adults consistently report a decline in core beneficial genera – including *Faecalibacterium*, *Coprococcus*, *Eubacterium*, and *Roseburia* – which produce anti-inflammatory short-chain fatty acids¹. Meanwhile, potentially pathogenic or pro-inflammatory microbes increase with age: for instance, *Ruminococcus gnavus*, *Eggerthella lenta*, and various *Clostridium* species become more abundant in the elderly and have been linked to frailty¹. “

“The oral microbiome also shifts discernibly with age, with implications for both oral and systemic health. Older adults typically experience a decline in dominant health-associated genera like *Streptococcus*, *Granulicatella*, and *Veillonella*, along with an increase in anaerobic taxa implicated in periodontitis and oral dysbiosis^{5,6}. “

Comment #5:

The introduction would benefit from stating that dysbiosis (changes to the microbiome) was recently added as a new hallmark of aging. <https://pubmed.ncbi.nlm.nih.gov/36599349/>

Response #5:

The inclusion of dysbiosis as a hallmark of aging is an interesting and relevant inclusion for the scope of work. We have included dysbiosis as a relevant topic in the discussion of chronological aging (Line 54), as well as dysbiosis throughout the discussion of body site specific microbiomes (lines 94-96). The suggested citation has also been included (reference 2).

“The human microbiome undergoes notable taxonomic and functional shifts as we age, influencing immune regulation, metabolism, dysbiosis, and disease risk^{1,2}.”

“ Older adults typically experience a decline in dominant health-associated genera like *Streptococcus*, *Granulicatella*, and *Veillonella*, along with an increase in anaerobic taxa implicated in periodontitis and oral dysbiosis^{5,6}.”

Comment #6:

In line 65, alpha and beta diversity should be briefly defined to ensure clarity. Moreover, as per my previous comments, the sentence "linked to chronological and biological aging." should be expanded. Was the property linked with accelerated aging?

Response #6:

We thank the reviewer for their feedback regarding the need to define alpha and beta diversity, and to add clarity around the implied link to accelerated aging by using more approachable language for clarity. The manuscript has been updated accordingly. Specifications are provided in the context of diversity to expand upon the literature beyond the changes in microbiome diversity (lines 62-65).

"Notably, these microbiome alterations correlate more with biological aging and frailty than with chronological age alone⁴, indicating that a healthy aging microbiome is characterized by specific taxonomic patterns rather than just overall diversity or load. "

Comment #7:

The paper would benefit from a brief introduction of the difference between chronological and biological aging.

Response #7:

We see the necessity to include definitions for chronological age and biological age, given the application of our healthy age prediction models to the exploration of biological age. The introduction of chronological age and biological age has been added to the background section of the paper (Lines 65-73).

"Chronological age simply counts years lived, whereas biological age, estimated from integrative molecular and clinical biomarkers, more accurately reflects functional decline and health risk⁷. Emerging microbiome clocks operationalize this concept by training machine-learning models on gut, skin, and oral community profiles to infer host age; the predicted value, often termed microbiome age, can deviate markedly from chronological age⁸. The magnitude and direction of this deviation correlate with phenotypes of healthy aging: older microbiome age ($\Delta\text{Age} > 0$) aligns with frailty and increased four-year mortality, whereas a younger-than-expected microbiome is linked to enhanced metabolic and inflammatory resilience^{4,9-11}. "

Comment #8:

Sentence in lines 73 and 74 "As individuals age, changes in the oral microbiota 74 mirror the physiological changes occurring in the host." should be supported by a reference.

Response #8:

We thank the reviewer for addressing this concern and have revised the manuscript accordingly. The aforementioned lines have been removed and replaced with the updated background section.

Comment #9:

As per my previous comments, the age-related changes to skin microbiome should be more detailed (taxonomy, load, increased, decreased, etc?)

Response #9:

We agree with the reviewer's concern that more information should be included to detail the age-related changes to skin microbiome and have revised the manuscript accordingly. Lines 480-492 include specifics for the taxonomy, load, and trends seen from the literature and analysis in terms of chronological aging.

"*Corynebacterium simulans* has been shown to be more abundant in adult skin than childhood skin, indicating a link with sebum secretion and a potential role in the aging process ⁴³. *Lactobacillus*, identified as a feature associated with younger individuals, has been linked to skin health through antimicrobial activity against skin pathogens, highlighting their potential role in maintaining skin health and reducing skin inflammation ⁴³⁻⁴⁵. *Cutibacterium* is also crucial for skin health, showing correlations with chronological aging and signs of aging ¹⁹. Specifically, *Cutibacterium acnes* is a dominant bacterium on human sebaceous skin, and particular clades are commonly associated with acne vulgaris ⁴⁶. *Cutibacterium* is most dominant during puberty on sebaceous skin sites (including face, scalp, upper back), and decreases in abundance as sebum secretions decrease with older age ⁴⁷. Its higher abundance in younger individuals makes it a well-known marker for host age ^{48,49}. However, its over-proliferation has been implicated in contributing to acne, and specific strains of *Cutibacterium acnes* dominate the skin microbiome of adult acne patients, indicating that its involvement in skin conditions may complicate the age signal ^{49,50}."

Comment #10:

Sentence in line 86 "Differential abundance analysis (...) should be supported by a reference.

Response #10:

We appreciate the reviewer's thoughtful recognition of opportunities to include supporting references. The manuscript has been revised accordingly (Line 119-122) with the inclusion of the reference "A multi-study analysis enables identification of potential microbial features associated with skin aging signs".

Comment #11:

Line 88 "(as measured by corneometer)" seems redundant

Response #11:

We appreciate and agree with the reviewer's comment and have removed the redundant portion of text.

Comment #12:

In line 100, "(RF, SVM)" should use full names, and state that they are ML methods.

Response #12:

We thank the reviewer for identifying this concern and have revised the line to expand all acronyms to their full names for the first occurrence of the acronym.

Comment #13:

In line 116 "Transformer models (...)" should include a reference, or if there is not one, should state potential directions of research, while highlighting that this kind of research has not been published yet.

Response #13:

We appreciate the reviewer's continued reflection to identify statements that would benefit from supportive citations. We have added references and discussion to highlight the model benefits, and existing literature to support these claims (Lines 204-218).

"Another emerging theme is the application of transformer-based architectures and large-scale pretraining for microbiome analytics²⁸. Transformer neural networks, which excel at modeling sequence and context due to the attention mechanism, are being adapted to microbial data to capture complex relationships across hundreds of taxa or genes. Transformer models can excel at microbiome analysis because they can attend to multiple microbial species simultaneously, identifying both obvious and hidden relationships between different taxa that might influence health outcomes, much like how understanding a conversation requires recognizing connections between words spoken at different times. Recent reviews highlight that transformer models can create rich, context-aware embeddings of microbiome profiles that outperform traditional models on downstream tasks²⁹. For example, the MetaTransformer model was used to integrate metagenomic and metabolomic data, yielding improved predictive power in microbiome studies²⁹. Transformers can attend to subtle co-occurrence patterns and long-range interactions (much like learning a language of microbial communities), which is promising for improving age predictions from high-dimensional metagenomic data. While still in early stages, such deep learning innovations are likely to enhance microbiome-based age predictors."

Comment #14:

Please double check your statement in line 128 "improves age prediction by >10% for all skin (...)" because in table 1 you have some lower improvements (e.g., 2% for 16S gut)

Response #14:

We revised to highlight that TRPCA remains comparable for all tasks, while performing better on other age prediction tasks. Additionally, we provided additional clarity in regard to the biological advancements of the work (Lines 228-241).

"This approach enables our model to capture contextual relationships within microbiome samples, similar to MDeep and phyLoSTM frameworks, while offering key architectural advantages: it eliminates overfitting risks during data compression and allows phylogenetically similar taxa to exhibit different feature importance directionalities. By assembling a comprehensive investigation of microbiome and age, with 16S rRNA sequence data from 8,959

samples from 10 studies and metagenomic data from 9,356 samples from 56 studies, we demonstrate that TRPCA is at least comparable to other popular machine learning models for all skin, oral, and gut microbiomes tested, regardless of sequencing method, while improving performance when trained via Multi-Task Learning (MTL). Furthermore, we describe novel methods to interpret the features from TRPCA on a global and sample level using SHapley Additive exPlanations (SHAP) values and RPCA feature loadings, leveraging the TRPCA architecture to compare the accuracy of age prediction across body site and sequencing method, uncover features driving the process of aging, and highlight the age prediction residual as a reproducible host associated attribute. “

Comment #15:

In line 163, RPCA drops "features with low prevalence" before RCLR, but the threshold for "low prevalence" is unspecified. Please specify for reproducibility

Response #15:

The method section lacked specification of the prevalence filtering, so we updated the methods section to include the filtering of features which occurred in less than 25 samples (Line 282-286).

“Microbiome datasets were preprocessed, dropping features with low prevalence (present in less than 25 samples) and log-transformed via RCLR. Dimensionality reduction was completed via PCA (scikit-learn version: 1.3.0).”

Comment #16:

The references don't follow a standardized way and are not formatted to journal guidelines (see at <https://www.nature.com/commsbio/submit/submission-guidelines#references>) Reference number 18 is not properly cited according to journal standards. Reference 36 is not properly cited. Moreover, since 36 is a preprint, it should either be replaced by a peer-reviewed article, or it should be noted in the manuscript that this reference was not supported by peer review. 58, 51, 47, 44 are also not properly referenced.

Response #16:

We appreciate the attention to detail for the references and have updated the references using the Nature style and replaced all preprints for peer-reviewed literature.

Comment #17:

The authors reference two models for DL for microbiome-derived disease prediction, however, it is not relevant to the scope of their research (age prediction). It is imperative that previous work in the field is appropriately referenced.

Response #17:

We appreciate the reviewer identifying the distinction between the models being referenced and the scope of this research. We have added additional references to address this concern. Additionally, the machine learning and deep learning section of the manuscript has been

updated to include more targeted discussion of the relevant machine learning literature (Lines 136-203) and how it supports age prediction modeling (lines 167-203).

“Microbiome clock models have emerged as specialized regressors for host age. These often integrate multiple data types and algorithms to boost accuracy. Initializing these efforts, in combination with deep learning approaches, an aging clock was demonstrated as the best approach for predicting host age from gut microbiome profiles, with a MAE of 5.9 years and providing a starting point for anti-aging intervention via feature importance⁸. Furthermore, the analysis provided foundations for feature extraction from deep learning models by evaluating the shift in model performance as a function of feature value perturbation⁸. Another example of leveraging machine learning for age prediction was an ensemble model developed using “multi-view” input, combining species abundance and pathway abundance profiles from gut metagenomes²⁶. The integrated model (an ensemble of heterogeneous learners) achieved high accuracy in predicting chronological age ($R^2 \approx 0.60$, MAE ~ 8.3 years) after accounting for geographic and technical confounders²⁶. The inclusion of functional data alongside taxonomy improved performance, underscoring that both community composition and its functional gene content change with age. Another advanced model called gAge introduced a composite biological age predictor based on the gut microbiota⁹. The gAge framework integrates gut microbial features “from different perspectives” (e.g. taxonomic, functional, and possibly metadata) using an ensemble of models, and notably it leverages unpaired samples to use more data⁹. This ensemble significantly improved prediction accuracy over any single model⁹. In an elderly cohort, gAge’s output and the deviation of predicted age from actual age (the residual) were strongly associated with health and frailty status⁹. This suggests microbiome-based age predictors can serve as clocks for biological age, where a microbiome “older” or “younger” than expected correlates with frailty or healthy aging, respectively.

Several studies have applied these models to various body sites and populations. Apart from adult gut clocks, researchers have built age predictors for early life and other niches. A XGBoost model was trained on healthy infant gut microbiomes to define a “microbiota age,” and a derived z-score was used to identify growth faltering²². In that work, specific bacteria (e.g. *Faecalibacterium* spp., *Bifidobacterium* spp.) and microbial pathways were identified as top contributors to age predictions, reflecting the orderly succession of the gut microbiome in infancy²². On the other end of the spectrum, studies of older adults indicate that chronological age itself may be less predictive within a narrow range of elderly years and, instead, microbial shifts often align more with biological aging or frailty. For instance, a longitudinal analysis found that microbiome composition in older adults tracked with frailty more than chronological age differences⁴. Together, these benchmark studies showcase the range of models (from random forests to deep neural networks) and use-cases (gut, oral, skin, infant, elderly) in microbiome age prediction, along with typical performance metrics. Currently, gut microbiome-based clocks for human chronological age can achieve roughly 5–10 year accuracy in adults²⁶, while specialized contexts (like infant growth or multi-omics clocks) can be even more precise^{22,27}.”

Comment #18:

The authors did not reference the seminal paper for DL for age prediction [https://www.cell.com/iscience/fulltext/S2589-0042\(20\)30384-9](https://www.cell.com/iscience/fulltext/S2589-0042(20)30384-9) which was the first research

attempt using the microbiome data and DL for age prediction. Moreover, this model is patented with a US patent

<https://patentimages.storage.googleapis.com/31/dc/2e/be5f9918f01378/WO2020084536A1.pdf>

Response #18:

Given the crucial importance of the inclusion of this work, we have included and thoroughly discussed the seminal paper on age prediction using DL (First mention line 167-171). Additionally, we include specific reference to the work in lines 606-618 in our discussion of previous work and aging clocks. We appreciate your highlighting of this gap in our written analysis, as this work is crucial for framing the progress and current state of age modeling via microbiome data.

“Our microbiome aging models demonstrate performance on par with, or exceeding, previously published microbial aging clocks. For example, Galkin et al. (2021) reported a deep learning model for gut microbiome age prediction with a MAE of ~5.9 years on an independent test set⁸. The finding in this study is significant, as the model performed well on data that was uncharacteristic and out of the training distribution, as well as on subjects that were donors with type 1 diabetes; however, as neural networks are continuous function approximators, well regularized models can generalize on unseen data. It should also be noted that the MAE metric of 5.9 years is based on a dataset with multiple samples from a given individual, and with a median age assignment error of 9.27 years. To give context for our WGS stool analysis, a median age assignment would yield an average error of 22.45 years; however, given differences in cohort size and composition, our WGS stool age predictions accuracies are consistent with the literature. As seen in other age prediction modeling, simpler models (e.g., RF) in prior studies often plateau at 7–10 years MAE for gut microbiomes⁸.”

Comment #19:

The paper would benefit from referencing this research as an example of previous work <https://academic.oup.com/bib/article/22/3/bbaa073/5835556#248048567>

Response #19:

We appreciate the suggestion to add “A novel deep learning method for predictive modeling of microbiome data,” as this method highlights a contextually aware deep learning approach for microbiome data. The reference has now been discussed and cited in the deep learning background (Lines 158-166). We appreciate the addition of this literature as we believe the results from the phylogenetically informed model support the hypothesis that hierarchical features within the context of a microbiome can improve model performance. Although TRPCA doesn’t explicitly encode phylogeny, the attention mechanism is keen on picking up nuanced patterns which may be occurring between orthogonal dimensions from the PCA reduced data.

“Similarly, the MDeep method leverages phylogenetic tree depth and deep learning to outperform competing methods for regression and binary classification²⁵. The hierarchical taxonomic convolutional layers mirror taxonomic levels in the phylogenetic tree, allowing for an

informative compression of OTUs before passing the representation to a fully connected, dense neural network– with the assumption that phylogenetically adjacent OTUs effect the target prediction in the same direction (i.e. one cluster of adjacent OTUs would be assumed to be only associated with older individuals). MDeep outperforming other neural network architectures, as well as Random Forest, demonstrates the importance of contextualizing the microbiome for improving model performance. “

Comment #20:

After introducing background evidence as mentioned above, the authors should briefly differentiate between the types of DL used above, and their model: how are they different (in a language that a physician/ not computational biologist would understand)

Response #20:

We appreciate the reviewer’s comments. We have addressed the comment by expanding the discussion surrounding the Deep learning foundational work that is now thoroughly discussed (Lines 144-214), with general descriptions provided to highlight key concepts for the deep learning model architectures. We have briefly outlined the advantages of the TRPCA model in contrast with other models (Lines 218-227). Below are a few examples:

“DeepMicro, is one such model using an ensemble of autoencoder-based pipelines to classify phenotypes from microbiome data²³. Autoencoder models learn a compressed, denoised representation of the data, isolating the most relevant features for representing the sample.”

“This CNN–LSTM approach extracts local taxonomic patterns via convolution and captures temporal dependencies via recurrent units, which could be adapted to improve age predictions in longitudinal aging cohorts (e.g. tracking an individual’s microbiome age trajectory).”

“Similarly, the MDeep method leverages phylogenetic tree depth and deep learning to outperform competing methods for regression and binary classification²⁵. The hierarchical taxonomic convolutional layers mirror taxonomic levels in the phylogenetic tree, allowing for an informative compression of OTUs before passing the representation to a fully connected, dense neural network– with the assumption that phylogenetically adjacent OTUs effect the target prediction in the same direction (i.e. one cluster of adjacent OTUs would be assumed to be only associated with older individuals)”

“Our model differs from previous deep learning approaches to microbiome analysis by applying transformer architecture and attention mechanisms (without pretraining) to learn patterns between orthogonal components derived from RPCA dimensionality reduction. This approach enables our model to capture contextual relationships within microbiome samples, similar to MDeep and phyLoSTM frameworks, while offering key architectural advantages: it eliminates overfitting risks during data compression and allows phylogenetically similar taxa to exhibit different feature importance directionalities.”

Comment #21:

In line 256, the sentence references a comparison to "other methods" but Figure 2 doesn't utilize them. In this case, studies with the "other methods" should be referenced, and the sentence would benefit the Discussion section, rather than results

Response #21:

The previous reference to "other methods" was specifically to the other model architectures in the TRPCA analysis. Line 379-381 now properly references the corresponding model comparisons:

"TRPCA improved MAE for host age prediction by up to 28% compared to the other model architectures (SVR, GBR, KNN, NN, and RF), demonstrating the greatest improvements in MAE for the skin microbiome samples (**Table 1, Figure 2, Table S2**)."

And the discussion now contains discussion about the model performance comparison in lines 624-629:

"From a statistical standpoint, we took measures to ensure the rigor of our performance estimates. We used hold-out validation and cross-validation to confirm that our MAE improvements were not overfitting. Additionally, model comparisons were made to highlight performance improvements for the various age prediction tasks. TRPCA significantly outperformed all models for 5 out of 15 total regression tasks, with the KNN and RF frequently performing on par with TRPCA (**Figure S23**)."

Comment #22:

Also in line 256, I would refrain from using the word "significant" unless you are referring to statistical significance

Response #22:

We thank the reviewer for identifying the potential misinterpretation of this word, and removed it in response to this suggestion.

Comment #23:

The MAE improvement percentages should be reported with standard deviations to show consistency across folds, if available

Response #23:

Table 1 has been updated to include the standard deviations across folds to show consistency across the folds. Table 2 has been updated to show the standard deviation for the MTL metrics across folds as well.

Comment #24:

Here is a tiny and petty detail, please remove the space in front of the full stop in line 285:)

Response #24:

We thank the reviewer for noticing this flaw and have removed the extra space.

Comment #25:

In lines 310-311 " *E. faecalis* was the most important feature for age prediction" please specify whether it was just the presence or abundance or other of this bacterium.

Response #25:

The manuscript text (lines 433-435) was updated to reflect that the importance was centered on the abundance of the taxa.

"However, for the first cluster of samples, *E. faecalis* abundance was the most important feature for age prediction, and other features had low feature importances."

Comment #26:

In line 321 "For example, for the age group 28-35, (...)" I believe this sentence is one of the clinical pearls of this research, and the features driving the age to older/younger should be discussed in more detail. I believe this finding is so good and so industry relevant that it requires description in detail and its own subsection. So far all the relevant information gets lost in the technical description of the model, and readers are likely to miss it, which would be a shame.

Response #26:

We appreciate the reviewer's perspective in highlighting the clinical importance of our findings in the 28-35 age group. The driving features described in Line 444 have been expanded upon in Biological and Technical Insights from Residual Age Predictions (Lines 578-585). We appreciate the feedback for this finding as it highlights the novel findings which can be extracted from the TRPCA interpretation method.

"Our analysis of model feature importance by age group revealed that certain microbial features disproportionately influenced model predictions, notably in individuals aged 28–35 for WGS skin data (**Figure 5a**). In this younger adult cohort, the model tended to predict lower ages as a function of a feature's abundance. SHAP values highlighted two organisms in particular, *Cutibacterium granulosum* and *Malassezia globosa*, as drivers of skewed age estimates. By examining SHAP values, we identified concrete microbial candidates that may serve as sentinels of accelerated or decelerated microbiome aging in an individual based on the influence of the feature to drive model prediction toward younger or older ages."

Comment #27:

I believe the paragraph about transformers lines 352-363 belongs in the introduction. If I was a non computationally trained reader, I would have largely benefited from this description before reading the results. In the first sentences of "discussion", however, I expect a clear and brief description of what was done and what it means, not the technical details. In reality, your discussion starts in line 365.

Response #27:

The text pertaining to the transformer models and architecture has been removed from the discussion and expanded upon in the deep learning background section. Lines 204-218 briefly describe the transformer model advantages in a generalized way, with citations for a few key publications that are suitable for the Communications Biology readership.

Comment #28:

Speaking of discussion, please highlight the finding of "divers" of skewed age prediction. I humbly believe this is one of your most important findings and definitely needs more attention.

Response #28:

Discussion connecting the age driver to the background literature has been included in the Biological and Technical Insights from Residual Age Predictions section. Specifically, drivers are linked to features documented in the literature in lines 586-605.

"Beyond these two taxa, our models identified a broader panel of age-associated microbes across skin, gut, and oral sites, each with potential impacts on host health. For instance, a relative abundance of certain oral anaerobes (e.g. *Fusobacterium*) emerged as markers of an older oral microbiome in our models, consistent with literature showing periodontal bacteria increase with age in some individuals⁶. These organisms are well-known contributors to gum disease, which tends to worsen with age and can have systemic inflammatory effects. Their prominence in age-prediction models suggests that oral health status is reflected in the microbiome age signal. Likewise, in the gut we noted that model coefficients and SHAP values often flagged depletion of beneficial genera (e.g. *Faecalibacterium*) and enrichment of pro-inflammatory taxa (e.g. *Escherichia/Shigella*) in individuals predicted to be older than their chronological years. The ability to predict host age from microbiome data carries several translational implications. In the realm of preventive and personalized medicine, a microbiome age model could be developed into a biological age indicator alongside other clocks (epigenetic, proteomic, etc.). There are also clear opportunities in the skincare and cosmetic industry. The skin microbiome was a strong component of our models, and prior work shows it to be a reliable mirror of chronological age^{19,21}. Probiotic supplementation is another direction as our findings identify specific taxa (*Faecalibacterium* in the gut, or *Veillonella* in the oral cavity) that decline with age⁶. Restoring fiber-degrading, butyrate-producing bacteria in an elderly gut microbiome might not only make the microbiome profile younger but also confer benefits like improved colonic health, reduced inflammation, and better metabolic regulation⁷⁹. "

Comment #29:

Line 378, as per my previous comments, what kind of correlation? Does the abundance of this bacterium drive or de-accelerate the rate of aging?

Response #29:

As the analysis was missing important details about the directionality of changes for specific taxa, we updated the text to include directionality associated with *Corynebacterium* and *Lactobacillus* for the microbiome skin section (lines 479-485).

"*Corynebacterium simulans* has been shown to be more abundant in adult skin than childhood skin, indicating a link with sebum secretion and a potential role in the aging process⁴³. *Lactobacillus*, identified as a feature associated with younger individuals, has been linked to skin health through antimicrobial activity against skin pathogens, highlighting their potential role in maintaining skin health and reducing skin inflammation⁴³⁻⁴⁵. *Cutibacterium* is also crucial for skin health, showing correlations with chronological aging and signs of aging¹⁹. "

Comment #30:

I believe your discussion of bacteria types and your findings should be restructured like "description of bacterium's one (specific!) implication in aging. Your model's findings about bacterium 1." then move to bacterium 2 and your findings about bacterium 2. In this way, you will avoid the feeling that it is an introduction again.

Response #30:

For the discussion on TRPCA identified microbial features, the Skin, Oral, and Gut microbiome sections (lines 474-538) have been restructured to pair the findings associated with the feature with the implication in health and aging. We believe this change improves the readability of the discussion section and makes logical sense for the introduction of taxa.

Comment #31:

For your discussion of gut microbiome (415-431) there is not a single reference to your findings. Please, change that.

Response #31:

Considering much of what you suggested about the previous mention of taxa, we have directly referenced our findings, as well as specify the directionality of the association with the taxa.

Comment #32:

Line 431, please modify "improves age prediction" to "improves age prediction accuracy" and state the model of comparison that indicates this improvement, and by how much. Same for 6% reduction in line 440: compared to what?

Response #32:

As the manuscript text was vague, we have updated the text to reference the model being used for comparison and specify that we are directly addressing age prediction accuracy. We have also updated the conclusions section to specify the comparison being made for prediction accuracy.

"Our study demonstrates that TRPCA significantly improves age prediction accuracy from microbiome profiles across various body sites and sequencing methods. We observed substantial reductions in MAE for age prediction, with improvements of 14% for skin (16S) and 28% for skin (WGS) compared to the best performing conventional ML methods (KNN for 16S and WGS skin samples, **Table 1**). "

Comment #33:

I believe the paragraph in lines 444-449 needs more attention to detail. Specifically, please differentiate between which bacteria are the drivers and which are the de-accelerators of the rate of aging, and whether you refer to their general abundance, detention, or other parameters.

Response #33:

In line with most of the taxa related statements for our findings, we have included explicit details of the relationship between the taxa's abundance and host age.

Comment #34:

Limitations belong in the discussion section.

Response #34:

Thank you for the recommendation, we agree that this section should be moved to the Discussion section. As the reference to limitations of the model were mentioned in the results and briefly in the methods, these comments were moved to the discussion.

Comment #35:

The publication would benefit from a brief sentence about the implications of the research. Is it an industry device potential? Clinical settings? How can it be useful outside of the lab?

Response #35:

The clinical and industry implications of the findings have been added in lines 586-605, through linking known microbiome aging markers with potential opportunities in health and wellness.

“The ability to predict host age from microbiome data carries several translational implications. In the realm of preventive and personalized medicine, a microbiome age model could be developed into a biological age indicator alongside other clocks (epigenetic, proteomic, etc.). There are also clear opportunities in the skincare and cosmetic industry. The skin microbiome was a strong component of our models, and prior work shows it to be a reliable mirror of chronological age^{19,21}. Probiotic supplementation is another direction as our findings identify specific taxa (*Faecalibacterium* in the gut, or *Veillonella* in the oral cavity) that decline with age⁶. Restoring fiber-degrading, butyrate-producing bacteria in an elderly gut microbiome might not only make the microbiome profile younger but also confer benefits like improved colonic health, reduced inflammation, and better metabolic regulation⁷⁹. “

Comment #36:

The discussion lacks MAE comparison to other papers using DL to measure age such as aforementioned [https://www.cell.com/iscience/fulltext/S2589-0042\(20\)30384-9](https://www.cell.com/iscience/fulltext/S2589-0042(20)30384-9) with MAE=5.9 based on microbiome taxonomic profiles (the comparison should be more detailed)

Response #36:

The discussion section has been expanded to include discussion of other models, including the work from Galkin et al. ([https://www.cell.com/iscience/fulltext/S2589-0042\(20\)30384-9](https://www.cell.com/iscience/fulltext/S2589-0042(20)30384-9)) in lines

606 to 617. The discussion also comments on the age ranges and sequencing methods used for our analysis, explaining some discrepancies between methods.

“Our microbiome aging models demonstrate performance on par with, or exceeding, previously published microbial aging clocks. For example, Galkin et al. (2021) reported a deep learning model for gut microbiome age prediction with a MAE of ~5.9 years on an independent test set⁸. The finding in this study is significant, as the model performed well on data that was uncharacteristic and out of the training distribution, as well as on subjects that were donors with type 1 diabetes; however, as neural networks are continuous function approximators, well regularized models can generalize on unseen data. It should also be noted that the MAE metric of 5.9 years is based on a dataset with multiple samples from a given individual, and with a median age assignment error of 9.27 years. To give context for our WGS stool analysis, a median age assignment would yield an average error of 22.45 years; however, given differences in cohort size and composition, our WGS stool age predictions accuracies are consistent with the literature.”

Comment #37:

Another limitation of the paper is the lack of biological age prediction, a more clinically relevant metric than chronological age, which could be noted as "further research"

Response #37:

We thank the reviewer for identifying this limitation, and have added to the Residual analysis Section to address this concern, as well as providing additional background to describe the difference between chronological and biological age. Finally, we have added biological age as a future direction for the research.

“In conclusion, TRPCA represents a significant advancement in microbiome data analysis, offering improved accuracy in age prediction and valuable insights into age-associated microbial features. This approach paves the way for a more nuanced understanding of the relationship between the microbiome and chronological aging, potentially informing future interventions aimed at promoting healthy aging through microbiome modulation and the exploration of biological age from a microbiome perspective. Our findings support the application of transformer-based methods in microbiome research and suggest promising avenues for the adoption of state-of-the-art model architectures in microbiome analysis.”

Comment #38:

The statistical analysis lacks rigor, omitting significance testing (e.g., t-tests) for MAE reductions—a practice increasingly expected despite being optional in some ML papers—while failing to address low R^2 values (e.g., 0.17 for 16S gut) that suggest poor fit and high variance in WGS skin (± 4.10 , $n=115$) that questions reliability. I suggest adding statistical tests, discussing R^2 implications, and exploring variance drivers (e.g., small sample size) to align with best practices and bolster claims.

Response #38:

We appreciate the reviewer's comments. T-test comparison of all regression tasks were added to supplemental. Discussion about correlation and other variance drives included in the Residual analysis section (lines 624-657). Please see Figure S23 and S24 for comparison of model performance and sensitivity analysis.

“From a statistical standpoint, we took measures to ensure the rigor of our performance estimates. We used hold-out validation and cross-validation to confirm that our MAE improvements were not overfitting. Additionally, model comparisons were made to highlight performance improvements for the various age prediction tasks. TRPCA significantly outperformed all models for 5 out of 15 total regression tasks, with the KNN and RF frequently performing on par with TRPCA (**Figure S23**). Still, we acknowledge that the absolute R² values are modest, especially for 16S stool samples (**Figure 2c, Figure 6a**); however, these findings are consistent across model architecture and two independent datasets, implying that 16S sequencing methods current do not resolve the human microbiome enough for aging studies. While TRPCA showed significant numerical improvements across datasets (**Figure S23**) with a 28% MAE reduction for skin WGS data (from 11.14 ± 2.65 to 8.03 ± 4.10), we acknowledge that overlapping uncertainty intervals in some cases indicate variable statistical significance of these improvements. These performance gains, though meaningful for computational modeling, likely offer limited additional clinical relevance compared to conventional ML approaches given our focus on chronological age in healthy individuals, rather than biological age between healthy and unhealthy cohorts. Nevertheless, the consistent performance pattern across body sites and sequencing methods reinforces TRPCA's value as a methodological advancement for microbiome-based age prediction research. Our sensitivity analysis of prediction residuals further validates the robustness of these findings. For many demographic variables, including sex and diet type, we observed no significant effects on age prediction residuals across most body sites and sequencing methods, suggesting our models are generally unbiased toward these factors. However, significant associations between residuals and certain variables were detected, particularly study-specific effects in 16S skin and oral samples and BMI effects in WGS samples (**Figure S24, Table S2-3**). These findings highlight potential methodological confounders and physiological factors that could influence microbiome-based age predictions. A notable limitation is the incomplete metadata across datasets, with variables such as non-Westernized status, diet, and BMI category frequently missing, preventing comprehensive assessment of all potential confounding factors across all body sites and sequencing methods. This metadata sparsity underscores the need for more complete demographic and lifestyle information in future microbiome aging studies to disentangle technical and biological influences on prediction accuracy. Given the results of the analysis, TRPCA is a promising tool for exploring the microbiome biological clock and has provided a substantial baseline for healthy aging. The analysis also highlights recommendations for methods and body sites which best support the investigation of the rate of aging.”

Comment #39:

I believe you have a typo in table 1 Skin (16S) stating your MAE as 5.09 ± 1.09 , but in Figure 2 you state it as 5.09 ± 1.07 .

Response #39:

The correct MAE is 5.09 +/- 1.07, and Table 1 has been updated to correct this mistake.